



# Response of global evaporation to major climate modes in historical and future CMIP5 simulations

Thanh Le[1, 2], Deg-Hyo Bae[1, 2]

[1]Department of Civil & Environmental Engineering, Sejong University, Seoul, South Korea
[2]Center for Climate Change Adaptation for Water Resources, Sejong University, Seoul, South Korea

*Correspondence to*: Deg-Hyo Bae (dhbae@sejong.ac.kr)

**Abstract.** Climate extremes, such as floods and droughts might have severe economic and societal impacts. Given the high costs associated with these events, developing early warning systems are of high priority. Evaporation, which is driven by around 50% of solar energy absorbed at surface of the Earth, is an important indicator of global water budget, monsoon precipitation, drought monitoring and hydrological cycle. Here we investigate the response of global evaporation to main modes of interannual climate variability, including the Indian Ocean Dipole (IOD), the North Atlantic Oscillation (NAO) and the El Niño-Southern Oscillation (ENSO). These climate modes may have influence on temperature, precipitation, soil moisture, wind speed, and are likely to have impacts on global evaporation. We utilized data of historical simulations and RCP8.5 future simulations derived from Coupled Model Intercomparison Project Phase 5 (CMIP5). Our results indicate that ENSO is an important driver of evaporation for many regions, especially the tropical Pacific. The significant IOD influence on evaporation is limited in western tropical Indian Ocean while NAO is more likely to have impacts on evaporation of the North Atlantic European areas. Land evaporation is found to be less sensitive to considered climate modes compared to oceanic evaporation. The spatial influence of major climate modes on global evaporation is slightly more significant for NAO and the IOD while slightly less significant for ENSO in the 1906-2000 period compared to the 2006-2100 period. This study allows us to obtain insight about the predictability of evaporation, and hence, may improve the early warning systems of climate extremes and water resources management.

**Keywords:** ENSO; IOD; NAO; Evaporation; CMIP5; Water resources management;

## 1 Introduction

The North Atlantic Oscillation (NAO; e.g., Hurrell et al., 2003), the Indian Ocean Dipole (IOD; Saji et al., 1999; Webster et
al., 1999), and the El Niño-Southern Oscillation (ENSO; e.g., Bjerknes, 1969; Neelin et al., 1998) are major modes of global climate variability. These climate modes may have influence on surface temperature (e.g., Arora et al., 2016; Leung & Zhou, 2016; Sun et al., 2016; Thirumalai et al., 2017; Wang et al., 2017), precipitation (Dai and Wigley, 2000), soil moisture (Nicolai-Shaw et al., 2016), humidity (Hegerl et al., 2015) and wind speed (Hurrell et al., 2003; Yeh et al., 2018), and are likely to have impacts on global evaporation and transpiration (hereafter simply referred as 'evaporation'). Evaporation,



which is driven by around 50% of solar energy absorbed at surface of the Earth (Cavusoglu et al., 2017; Jung et al., 2010), is an important indicator of global water budget, monsoon precipitation, drought monitoring and hydrological cycle (Friedrich et al., 2018; Kitoh, 2016; Lee et al., 2019; Van Osnabrugge et al., 2019; Son and Bae, 2015). Additionally, changes in global evaporation are expected to feedback on global and regional climate. For example, land evaporation is shown to have influences on carbon cycles (Cheng et al., 2017), cloud cover (Teuling et al., 2017) and air temperature (Miralles et al.,

35  2012).

While previous studies emphasize the importance of ENSO on global evaporation (e.g., Miralles et al., 2013), the roles of the IOD and NAO remain elusive. In addition, the future influence of these climate modes on global evaporation under warming environment remain unclear. Climate change and rising temperature might increase surface evaporation (Miralles et al., 2013), and thus, might reduce global water availability and cause change in hydrological cycle (e.g., Naumann et al., 2018).

Moreover, previous works mainly address the connection between individual climate mode and evaporation, however, the role of other climate modes might not be included in the analyses. As long term and reliable evaporation data is lacking (e.g., Hegerl et al., 2015; Miralles et al., 2016), climate model simulations provide additional opportunity to examine the impacts of main climate modes on global evaporation. Besides, evaluating the models' consistency in reproducing the impacts of internal climate variability on evaporation is important for understanding the difference between models.

Here we investigate the causal impacts of major climate modes (i.e. ENSO, IOD and NAO) on global terrestrial and oceanic evaporation in CMIP5 model simulations for the 1906-2000 and the 2006-2100 periods. For this investigation, we use multivariate predictive models and tests of Granger causality which consider the simultaneous impact of climate modes on global evaporation (see Methods section 2.2).

## 2      Data and Methods

### 2.1      Data

The data used in the present study was obtained from Coupled Model Intercomparison Project Phase 5 (CMIP5). We employ data of historical simulations (experiment name 'historical' in CMIP5) and future simulations with high-emissions climate change scenario (experiment name Representative Concentration Pathway [RCP] 8.5 scenario, 'rcp85') (Taylor et al., 2012). The starting year of historical simulations is roughly 1850 and the ending year is roughly 2005 while the starting year of

future simulations is roughly 2006 and the ending year is roughly 2100. In our analyses, we only use the data for the 1906-2000 historical period as a reference for the future period 2006-2100. Using other data period with similar length (i.e., 95 years) does not alter the results and conclusions. We use different data variables, including monthly sea level pressure (i.e., 'psl' in CMIP5 datasets), sea surface temperature (i.e., 'ts'), and evaporation (i.e., 'evspsbl'). For each model, we only utilize one simulation (i.e., 'r1i1p1'). The models employed in this study are listed in Table S1 (Supplement). Terrestrial

evaporation is the flux of water at surface into the atmosphere due to transformation of both solid and liquid phases to vapor





(from vegetation and underlying surface). Most of climate models do not provide separately the data of evaporation from canopy (i.e. transpiration) and water evaporation from soil (i.e. evaporation).

There might exist model biases in simulating ENSO (e.g. Taschetto et al., 2014), the IOD (Chu et al., 2014; Weller and Cai, 2013), NAO (Gong et al., 2017; Lee et al., 2018) and there is uncertainty in capability of land surface models in modeling

evaporation (Mueller & Seneviratne, 2014; Wang & Dickinson, 2012). However, CMIP5 data is useful and provides additional understanding about the connections between major climate modes and global evaporation.

## 2.2    Methods

The NAO index (Hurrell et al., 2003) is computed as the first empirical orthogonal function (EOF) of boreal winter (DJF) sea level pressure (SLP) anomalies in the North Atlantic area (90ºW-40ºE, 20º-70ºN). We compute the dipole mode index

(DMI) (Saji et al., 1999) as the discrepancy of SST anomalies between the western tropical Indian Ocean (50ºE-70ºE; 10ºN-10ºS) and the south-eastern tropical Indian Ocean (90ºE-110ºE; 0ºN-10ºS) in the boreal fall (SON). We define the ENSO index as the average sea surface temperature (SST) anomalies in the Niño 3.4 region (120°W-170°W; 5Nº-5ºS) during boreal winter. We include ENSO, IOD and NAO in our analyses as these indices are three major climate modes of tropical Pacific, tropical Indian and North Atlantic Oceans.

We evaluate the causal effects of a climate mode (i.e., NAO or DMI or ENSO) on evaporation by using the following predictive model (e.g., Mosedale et al., 2006):

$$X_t = \sum_{i=1}^{p} \alpha_i X_{t-i} + \sum_{i=1}^{p} \beta_i Y_{t-i} + \sum_{j=1}^{m} \sum_{i=1}^{p} \delta_{j,i} Z_{j,t-i} + \varepsilon_t \qquad \textbf{(1)}$$

where $X_t$ is the annual (or seasonal) evaporation for year $t$, $Y_t$ is the selected index (i.e., ENSO or NAO or DMI) for evaluating the causal effects on evaporation for year $t$, $Z_{j,t}$ is the confounding variable $j$ for year $t$, $p \geq 1$ is the order of the

causal model, and $m$ is total number of confounding variables. The optimal orders are normally less than 8 in our analysis, suggesting that the impact of major climate modes on evaporation is evaluated at inter-annual time scales. Confounding variables (e.g., if NAO is the selected index of possible causal influence on evaporation, the confounding variables are DMI and ENSO) may have impacts on the links between selected index and global evaporation. There are two form of confounding variables in our analysis, hence, $m$ is equal to 2 in equation (1). The regression coefficients $\alpha_i$, $\beta_i$ and $\delta_{j,i}$ and the

noise residuals $\varepsilon_t$ are computed by using multiple linear regression analysis and least squares method. All climate indices and evaporation data are normalized and detrended.

We apply test of Granger causality for the predictive model described in equation (1). Specifically, in order to assess the causal influence from $Y$ to $X$, we compute the probability of the null hypothesis for an absence of Granger causality from $Y$ to $X$. Additional information on the test of Granger causality are explained in earlier works (e.g., Le, 2015; Le et al., 2016).

The techniques employed in the present study provide robust assessment about the causal influence of considered climate





mode on global evaporation. In addition, these approaches account for the concurrent influence of confounding variables and hence, provide more realistic evidence of the response of global evaporation to major climate modes.

## 3    Results and discussions

**ENSO influence on evaporation**

The probability maps of no Granger causality between ENSO and global evaporation for the periods 1906-2000 and 2006-2100 are shown in Figure 1. In both periods, ENSO is more likely to have influence on evaporation of different regions (highlighted in blue shades) in both hemispheres, including middle Asia (regions closed to Caspian Sea, details are shown in Figure S1a in the Supplement), Indian Oceans, Indochina Peninsula, Australia (Figure S1b), tropical Pacific and northeastern South America (i.e. Amazonia, Figure S1c) and the Pacific coast of America (Figure S1d). There is high agreement between

models (indicated by stippling in Figure 1) in simulating ENSO-evaporation connection of these regions. Specifically, high agreement of climate models in teleconnection between ENSO and tropical oceans evaporation implies that models can simulate the impacts of ENSO on evaporation. ENSO might indirectly influence global evaporation by modulating regional climate factors associated with evaporation processes. For example, ENSO significantly influence near surface wind, which is the main contributor of change in evaporation (Xing et al., 2016). The ENSO impacts on large part of tropical Pacific

Ocean are robust at 5% and 10% significance levels (here we reject the null hypothesis of the absence of Granger causal effects between ENSO and evaporation at 5% and 10% significance levels, hence, we  conclude that there is significant causal impact; we note that the 5% and 10% significance levels are computed from the test for the absence of Granger causality). In the tropical Pacific, the influence of ENSO on evaporation might be associated with Wind-Evaporation SST (WES) effect (Cai et al., 2019). The WES effect occurs when warm (cold) water becomes warmer (colder) due to decrease

(increase) in evaporation and weakened (strengthened) surface winds. Besides, ENSO is known to induce changes in global precipitation with decrease in rainfall in Africa, Indochina Peninsula, Indonesia, Australia and Amazonia during El Niño phase (Dai and Wigley, 2000) and thus indirectly influence evaporation of these regions.

Additional analyses show a more robust influence from ENSO on global evaporation in boreal spring compared to other seasons (Figure S2 in the Supplement). The ENSO influence on boreal winter evaporation is mainly found in the tropical

Pacific, while land evaporation is less likely to be influenced by ENSO in winter (Figure S2a). The limited response of Northern Hemisphere boreal winter evaporation to ENSO might be related to low energy supply (i.e. incoming solar radiation), which leads to reduction in land evaporation (Martens et al., 2018). Although ENSO strength peaks in boreal winter, the weak response of evaporation to ENSO during this specific time of the year suggests the important role of other internal climate modes or external factors (i.e. solar radiation). The boreal winter evaporation in the Southern Hemisphere

might be controlled by local background conditions (e.g., surface temperature) and other major climate modes (e.g. the Southern Annular Mode [SAM]). The response of global evaporation to ENSO is found to be the most robust in boreal spring and gradually decrease in summer, fall and winter (Figure S2 and S3). These results indicate the persistent and lagged



influence of ENSO on regional evaporation (e.g., Australia, northeastern South America and tropical Pacific are influenced during boreal spring, summer and fall). The seasonal connection between ENSO and global evaporation in future simulations (Figure S3) shows similar patterns compared to historical simulations (Figure S2).

**IOD influence on evaporation**

Figure 2 describes the evaporation response to the IOD which is mainly found in Indian ocean and the tropical Pacific. The IOD impacts are shown to be significant in the western tropical Indian ocean close to the eastern coast of Africa (Figures 2a and 2b, see Figure S4a for additional details). The evaporation response in the western tropical Indian ocean to the IOD may be associated with the impact of the IOD on short rains in East Africa (Behera et al., 2006; Black et al., 2003). The IOD signature is also found in the areas close to the Horn of Africa (Figure 2a). This result is in agreement with previous work (Martens et al., 2018) which proposed a possible evaporation response of Horn of Africa to positive phase of the IOD. The IOD shows remote control of evaporation in parts of tropical Pacific, especially the eastern regions (Figure S4b). This teleconnection might be associated with the IOD effects on ENSO and SST in the eastern parts of tropical Pacific (e.g., Izumo et al., 2010; Le & Bae, 2019). In historical simulations (Figure 2a), the IOD impacts might reach as far as the Southern Ocean (region close to 150°W to 120°W; 45°S to 60°S) where there is high agreement between models. This IOD-Southern Ocean teleconnection might be indirect and is possibly related to other major climate mode of the Southern Hemisphere (e.g. the SAM).

Although the IOD is shown to contribute to droughts in East Asia (Kripalani et al., 2009) and Australia (Ashok et al., 2003; Cai et al., 2009; Ummenhofer et al., 2009), our analyses imply that the IOD impacts on evaporation of these regions are unclear, particularly in the future period of 2006-2100 (Figures 2b, S5 and S6). Additionally, there is uncertainty in the impact of the IOD on evaporation in other regions, including the middle East, South Asia, Southeast Asia where there is complex interactions between different climate modes (e.g., ENSO, IOD and the Indian summer monsoon rainfall; Cai et al., 2011; Le & Bae, 2019). Unlike seasonal responses of evaporation to ENSO, the seasonal responses of evaporation to the IOD indicate similar patterns between different times of the year (Figure S3 and S4). Specifically, in all four seasons, the IOD influence on evaporation is mainly shown in parts of tropical Indian and Pacific oceans. Although there is still uncertainty, IOD signal is found in evaporation change of Amazonia (i.e. northeastern South America, Figures S5a and S6a) in boreal winter (December–February) for several model simulations. These results are in agreement with previous study (Martens et al., 2018) which showed the sensitivity of evaporation in the rainforest to the IOD.

**NAO influence on evaporation**

The global evaporation response to NAO, which is limited in the Northern Hemisphere, is shown in Figure 3. NAO mainly contributes to change in evaporation of the North Atlantic European sector where high agreement between models is found (see Figure S7 for additional details). This conclusion shows consistency with earlier works and indicates the capability of models in simulating the connection between NAO and regional evaporation. Particularly, the positive phase of NAO leads



to changes in temperature, transport of atmospheric moisture and precipitation in northern, central and western Europe and parts of southern Europe and thus causes change in evaporation (Hurrell et al., 2003). Given significant economic losses from floods in Europe caused by NAO (Hurrell et al., 2003; Zanardo et al., 2019), the predictability of regional evaporation using NAO index might be potentially helpful to mitigate the flood impacts. There is uncertainty of NAO impact on southern North Atlantic and eastern Europe. In several models, NAO impact is found in small areas of the eastern tropical Pacific and

South Atlantic (Figure 3a). NAO might also influence evaporation in parts of central North America and the coast of northeastern South America; however, these signatures are unclear (Figure 3a). The sensitivity of evaporation in the western coast of North America shown in Figure 3a is somewhat in agreement with the findings of previous study (Martens et al., 2018).

The seasonal evaporation response to NAO is much weaker in boreal fall and winter compared to spring and summer (Figure

S8 and S9). In the Northern Hemisphere, this result might be due to decrease in seasonal solar radiation. The NAO impacts on northwestern Europe in boreal spring (Figure S8b and S9b) suggests several months lagged effect of NAO on evaporation of this region. Global evaporation response to NAO is the most robust in boreal spring where NAO signature might exist in large area of northern Eurasian continent and parts of Pacific Ocean (Figure S8b).

**Comparing the impacts of different climate modes**

Figure 4 shows the difference in multi-model mean probability for the absence of Granger causality between a pair of climate modes and annual mean evaporation. Specifically, the effects of ENSO on evaporation are more significant compared to NAO in large parts of tropical region (highlighted in blue shades which indicate lower probability of no causal impacts) while NAO effects are more significant in the high latitude region of Northern Hemisphere (highlighted in red shades), especially regarding the North Atlantic European sector (Figures 4a and 4d). We observe stronger signature of

ENSO compared to the IOD in the Middle East, the Pacific coast of North America and large parts of the tropics, except for the western tropical Indian Ocean (Figures 4b and 4e). Figure 4e indicates the increase in ENSO spatial impact over the western Indian Ocean in the 21$^{st}$ century compared to 20$^{th}$ century. This increase in ENSO impacts might be related to the increase in future extreme ENSO events (Cai et al., 2015). Interestingly, the IOD effects is found slightly stronger compared to ENSO in the North Pacific and North Atlantic, suggesting the potential role of the IOD in these regions. The impacts of

NAO are more significant compared to the IOD in the North Atlantic European sector while IOD impacts are stronger in the tropical Pacific and Indian Oceans and high latitude region of southern hemisphere (Figures 4c and 4f). Overall, ENSO is the dominant mode of tropical evaporation while NAO largely contribute to regional evaporation in the high latitude of Northern Hemisphere.

**Discussions**

The map of ENSO-evaporation connection presented here (Figures 1, S1, S2 and S3) confirm the results obtained previously (e.g., ENSO influence on evaporation of Australia and Amazonia where there is high consistency between models as shown





in Figure 1). The results shown here are partly in agreement with previous study (Miralles et al., 2013) which showed lower evaporation during El Niño events due to decrease in precipitation in eastern and central Australia and eastern South America. In addition, the robust signature of ENSO on evaporation of tropical regions is consistent with the findings of

Miralles et al. (2013) which showed negative (positive) anomalies of evaporation for most of the tropics under El Niño (La Niña) conditions. Besides, our results also show new features of the connection between ENSO and global evaporation. Specifically, ENSO is more likely to have influence on the Pacific coasts of both North and South America. The IOD is suggested to be the main climate mode to have impacts on evaporation of both hemispheres (Martens et al., 2018), however, our results indicate that ENSO influence also has similar characteristics (Figures 1, S1 and S2). In addition, there is

uncertainty of ENSO impacts on land evaporation, especially regarding the regions of South Asia, Africa and Southern South America. The result of ENSO influence on eastern Australia shows consistency with past findings (Martens et al., 2018; Miralles et al., 2013), however, we further indicate that there is also close connection between western Australia evaporation and ENSO variations (Figures 1, S1 and S2). In fact, evaporation in Australian continent was shown to have highest sensitivity to ENSO conditions compared to other continents (Miralles et al., 2013). Additionally, our results suggest

that ENSO is more likely to have impacts on evaporation of Australia during both historical period of 1906-2000 and future period 2006-2100. We note that in the present study, we use longer data periods (i.e. 1906-2000 and 2006-2100 model simulations) compared to recent works [e.g., 1982-2012 (Martens et al., 2018) and 1980-2011 (Miralles et al., 2013) observations]. Hence, the length of data period might affect the statistical significance tests and the interpretation of results. There are different factors that might contribute to the ambiguity of climate mode impacts on evaporation of several regions

(e.g. South Asia, Africa and Southern South America). Specifically, these factors include the large discrepancies of current estimations of land evaporation for recent decades (Dong & Dai, 2017; Miralles et al., 2016), the limitations of climate models in simulating climate modes (Gong et al., 2017; Lee et al., 2018; Taschetto et al., 2014; Weller and Cai, 2013) and the overestimation of simulated evaporation in most regions (Mueller and Seneviratne, 2014).

Figure 5 shows the fraction area of Earth surface for land and ocean with probability for the absence of Granger causality between climate modes and evaporation less than 0.1 (i.e., p value < 0.1; we note that the fraction area is substantially

smaller if p value < 0.05). Specifically, during the period 1906-2000, nearly 1.039% of land area is affected by ENSO at 10% significance level while the affected land areas by NAO and the IOD are 0% and 0.017%, respectively (Figure 5a). The area of oceanic evaporation influenced by ENSO, NAO and the IOD are 2.908%, 0.01% and 0.196%, respectively (Figure 5b). We observe an increase in land area affected by ENSO to 1.38% during the 2006-2100 period while the affected land

areas by NAO and the IOD are 0% (Figure 5c). The area of oceanic evaporation influenced by ENSO, NAO and the IOD are 2.944%, 0.003% and 0.122%, respectively (Figure 5d). This result shows a minor decrease in NAO and IOD effects on oceanic evaporation during the 2006-2100 period compared to the 1906-2000 period. Figure S10 shows additional analyses for the fraction area of Earth surface for land and ocean with probability for the absence of Granger causality between climate modes and evaporation less than 0.25 (i.e., climate modes are unlikely to have no causal effects on evaporation;

Stocker et al., 2013).



The considered climate modes (i.e. ENSO, NAO and IOD) are more likely to have influences on global evaporation over oceans while they have limited signature in change of land evaporation for many regions (Figures 1, 2, 3, 5 and S10). These results indicate the role of other factors in modulating land evaporation. Particularly, the influence of major climate modes on land evaporation might be offset by other factors like greenhouse gases, aerosols or solar radiation (Dong and Dai, 2017;

Hegerl et al., 2015; Liu et al., 2011). The impacts of climate modes on ocean evaporation contribute to change in global hydrological cycle as ocean evaporation might affect land water cycle by inducing change in regional precipitation (Diawara et al., 2016). For example, the evaporation of the eastern North Pacific is the main moisture supply for precipitation in California (Wei et al., 2016).

Changes in the spatial influences of major modes of climate variability on regional evaporation for future period 2006-2100

and historical period 1906-2000 depend on each climate mode (Figures 1, 2, 3). Analyses in details of the difference between these two periods are shown in Figure 6. Specifically, the fraction area of Earth surface showing lower probability of ENSO effects for 2006-2100 period is approximately 52.9% (Figure 6a). This result indicates that ENSO slightly expand the impacted regions (highlighted in red shades, Figure 6a) during 2006-2100 period compared to 1906-2000 period. Conversely, the fraction area of Earth surface for effects of NAO and the IOD during the 2006-2100 period are decreased

with 47.2% and 45.7%, respectively (Figure 6b and 6c). These results suggest that, for several regions of declining impacts of climate modes (highlighted in blue shades, Figure 6), evaporation processes in the 21$^{st}$ century tend to involve more local climate background conditions (e.g., precipitation, near-surface air temperature, wind speed, soil moisture). For example, response of regional evaporation to climate warming depends on precipitation (Parr et al., 2016; Zhang et al., 2018) and projected rise of surface temperature is shown to mainly contribute to the increase in regional evaporation (Laîné et al.,

2014). Because the volume of moisture carried by air increases with air temperature, the atmospheric water vapor demand is expected to increase with rising air temperature and rising greenhouse gases concentration (Miralles et al., 2013). In addition, the declines in pan evaporation in southern/western Australia are mainly caused by decreases in wind speeds (Stephens et al., 2018).

## 4   Conclusions

The CMIP5 historical and RCP8.5 future simulations provide an opportunity to assess the influence of major climate modes on global evaporation, which plays an important role in hydrological cycle, drought monitoring and water resources management. This paper employed tests of Granger causality and showed vigorous evaluation of possible impacts of NAO, the IOD and ENSO on global evaporation.

The results show that ENSO is likely to have impacts on evaporation of different regions in both hemispheres, including

tropical Pacific and Indian Oceans, Indochina Peninsula, middle Asia (regions closed to Caspian Sea), Australia, northeastern South America (i.e. Amazonia) and the Pacific coast of North and South America. The impacts of NAO are mainly found in the North Atlantic and European regions while the notable influence of the IOD is limited in western



tropical Indian Ocean and part of eastern tropical Pacific. Despite more extreme IOD events are expected in the future (Cai et al., 2013, 2014), the spatial influences of the IOD on evaporation are slightly less significant in the 2006-2100 period compared to the 1906-2000 period. The weak impacts of ENSO, NAO and the IOD on evaporation of several regions suggest the importance of external forcings (e.g. greenhouse gases radiative forcing, solar forcing) and other climate modes on global evaporation variability. We emphasize the strong connection between considered climate modes (i.e. ENSO, IOD and NAO) and oceanic evaporation at interannual time scales. Land evaporation is shown to have weak connection with teleconnection indices in several regions, suggesting the important role of local wind speed (Stephens et al., 2018), surface temperature (Laîné et al., 2014; Miralles et al., 2013), moisture supply (Jung et al., 2010) and amount of precipitation (Parr et al., 2016) on changes in land evaporation.

Our results may have suggestions for the predictability of regional evaporation (e.g. Australia, tropical Pacific, tropical Indian and North Atlantic Oceans, the Pacific coast of North and South America, Amazonia, Europe, Indochina Peninsula and middle Asia) by using past time series of major climate modes for short term of several years. The results of this study might provide information for drought and flood prediction as evaporation is an important metric for quantifying drought (McEvoy et al., 2016) and flood events.

Uncertainty regarding the impact of major climate modes on evaporation of several regions (e.g. ENSO impacts on evaporation of South Asia, South Africa, eastern North America, southern South America; the IOD impacts on western Africa and South Asia; NAO impacts on North Atlantic and surrounding areas) suggests that additional works are necessary. Further investigation about the effects of other internal climate modes (e.g., the Southern Annular Mode [SAM], the Indian Ocean Basin [IOB] mode, the North Tropical Atlantic [NTA]) on evaporation might improve our understanding of the response global hydrological cycle to internal climate variability. Little effort has been made to quantify the influences of external climate factors (i.e., volcanic eruptions, solar variations, and changes in concentration of greenhouse gases) on global evaporation, thus, these analyses might be a subject of forthcoming studies.

**Data availability**

CMIP data can be downloaded from the ESGF website at https://esgf-node. llnl.gov/projects/esgf-llnl/.

**Competing interests**

The authors declare that they have no conflict of interest.



## Author contribution

TL designed the study. TL performed the data analysis and wrote the manuscript. DHB contributed to the interpretation of results and the writing of manuscript.

## Acknowledgements

We thank Ryan Teuling for valuable comments and suggestions. We acknowledge the World Climate Research Programme's Working Group on Coupled Modeling, which is responsible for CMIP, and we thank the climate modeling groups (listed in 285 Table S1 of this paper) for producing and making available their model output. For CMIP the U.S. Department of Energy's Program for Climate Model Diagnosis and Intercomparison provided coordinating support and led the development of software infrastructure in partnership with the Global Organization for Earth System Science Portals. T. Le is supported by a research grant from Sejong University. This work is supported by the Korea Environmental Industry and Technology Institute (KEITI); the research grant is funded by the Ministry of Environment (grant RE201901084).

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



**Figure 1.** Multi-model mean probability map for the absence of Granger causality between ENSO and annual mean evaporation for the periods 1906-2000 (a) and 2006-2100 (b). Stippling demonstrates that at least 70% of models show agreement on the mean probability of all models at given grid point. An individual model's agreement is determined when the difference between the multi-model mean probability and the selected model's probability is less than one standard deviation of multi-model mean probability. The yellow (red) contour line designates $p$ value = 0.1 (0.05). Blue shades indicate low probability for the absence of Granger causality. ENSO = El Niño–Southern Oscillation.





MODELS MEAN: IOD - EVAPORATION PERIOD 1906-2000

MODELS MEAN: IOD - EVAPORATION PERIOD 2006-2100

**Figure 2.** As in Figure 1, but for Granger causality between IOD and annual mean evaporation. IOD = Indian Ocean Dipole.





**Figure 3.** As in Figure 1, but for Granger causality between NAO and annual mean evaporation. NAO = North Atlantic Oscillation.



**Figure 4.** Difference in multi-model mean probability for the absence of Granger causality between a pair of climate modes and annual mean evaporation. The results are shown for the periods 1906-2000 (a, b, c) and 2006-2100 (d, e, f). ENSO minus NAO (a, d). ENSO minus IOD (b, e). NAO minus IOD (c, f). Blue shades indicate lower probability for the absence of Granger causality. ENSO = El Niño–Southern Oscillation. NAO = North Atlantic Oscillation. IOD = Indian Ocean Dipole.





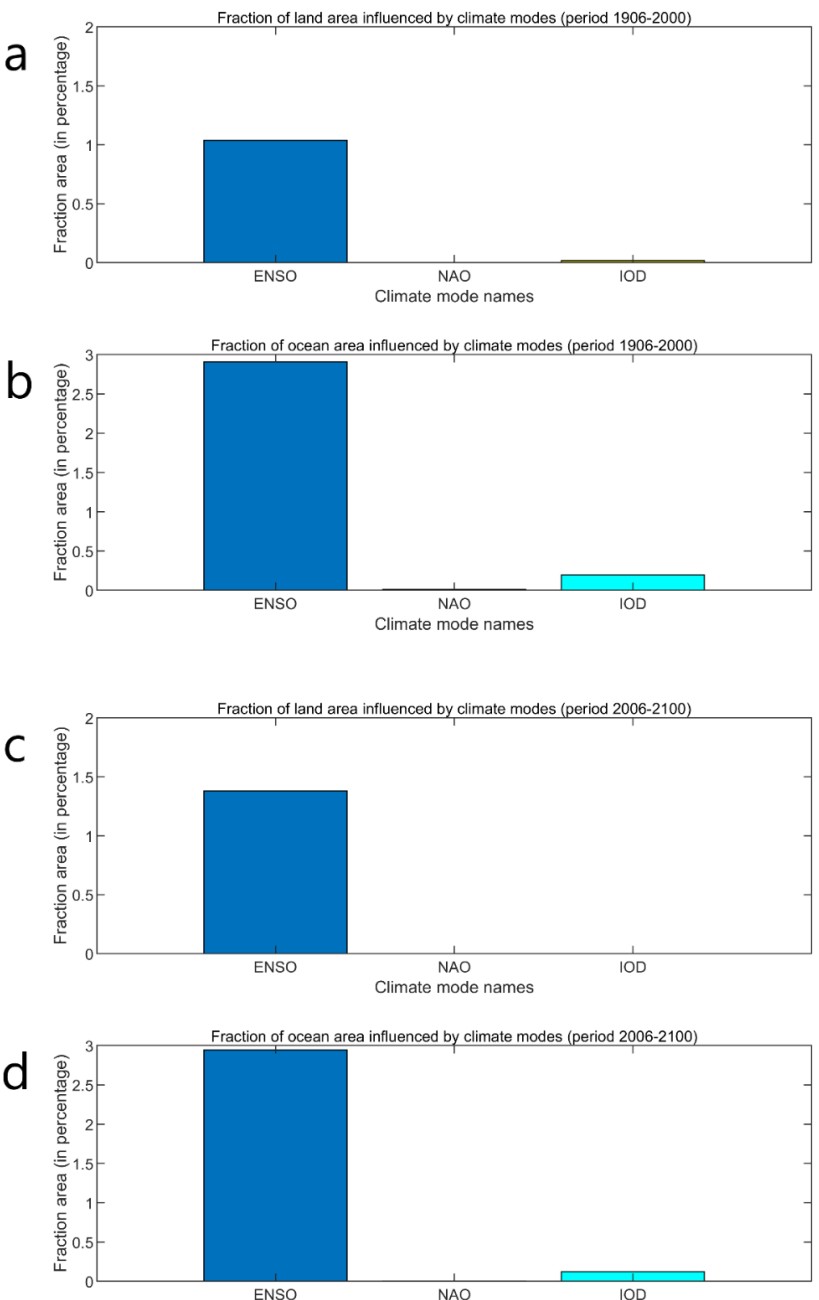

**Figure 5.** Fraction of Earth surface for land (a, c) and ocean (b, d) with probability for the absence of Granger causality between climate
modes and evaporation less than 0.1 (i.e., $p$ value $< 0.1$). The results are shown for the influence of individual climate mode on annual
mean evaporation for periods 1906-2000 (a, b) and 2006-2100 (c, d). Fraction area lower than 0.05% is plotted in yellow bar. Fraction area
higher than 0.05% and lower than 0.5% is plotted in cyan bar. ENSO = El Niño–Southern Oscillation. NAO = North Atlantic Oscillation.
IOD = Indian Ocean Dipole.





MODELS MEAN OF ENSO - EVAPORATION: PERIOD 1906-2000 MINUS PERIOD 2006-2100

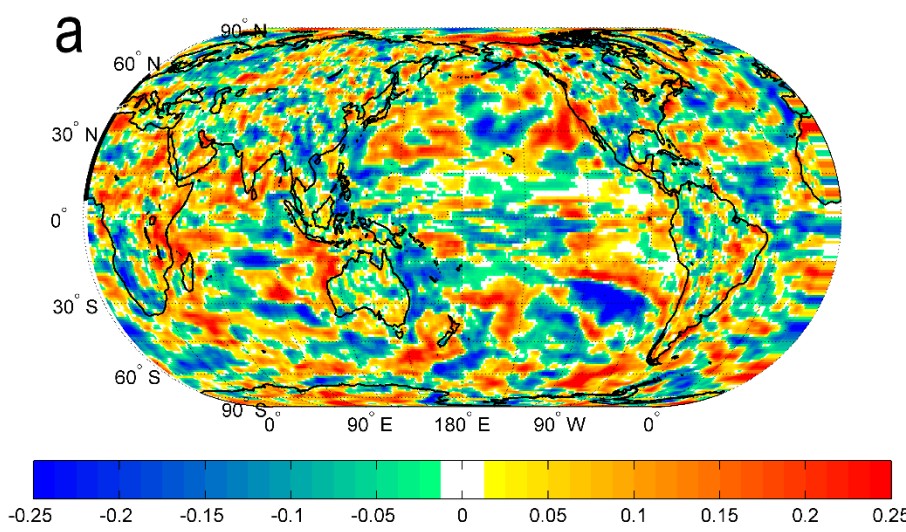

MODELS MEAN OF IOD - EVAPORATION: PERIOD 1906-2000 MINUS PERIOD 2006-2100

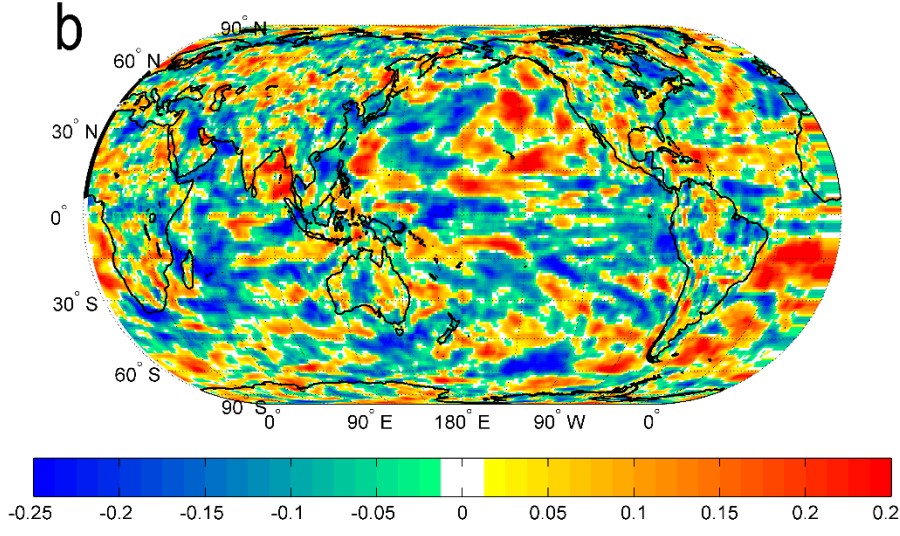

MODELS MEAN OF NAO - EVAPORATION: PERIOD 1906-2000 MINUS PERIOD 2006-2100

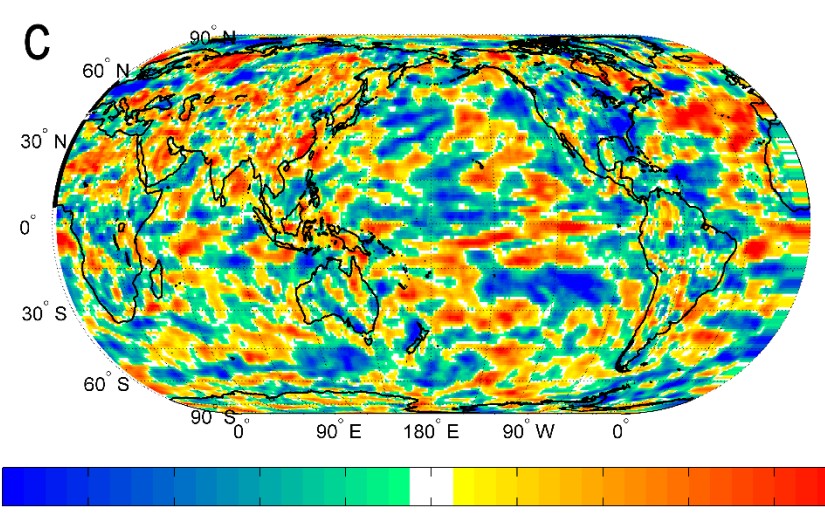



**Figure 6.** Difference of probability for the absence of Granger causality of individual climate mode on annual mean evaporation between periods 1906-2000 and 2006-2100 (i.e., period 1906-2000 minus period 2006-2100). The results are shown for ENSO (a), IOD (b) and NAO (c). Blue shades indicate lower probability of no Granger causality during period 1906-2000 compared to period 2006-2100. Red shades indicate lower probability of no Granger causality during period 2006-2100 compared to period 1906-2000. ENSO = El Niño–Southern Oscillation. NAO = North Atlantic Oscillation. IOD = Indian Ocean Dipole.
