# Peer review of "Response of global evaporation to major climate modes in historical and future CMIP5 simulations"

_Hydrology and Earth System Sciences, 2019_

## Referee Comment (RC1) · Brecht Martens (Referee) · 27 Sep 2019

**Le *et al.* (2019):** Response of global evaporation to major climate modes in historical and future CMIP5 simulations

**GENERAL COMMENTS**

This paper explores the historical and future impact of three major modes of internal climate variability on evaporation from oceans and land into the atmosphere based on data from CMIP5 model simulations and a Granger causality framework. Such an analysis might provide useful insights about the distribution of water resources in the near future and to help better forecast extreme hydrological events. As such, I truly see scientific value in this study; however, in my opinion, the present paper first needs to be improved in two ways: (1) the description of the method needs to be improved and more details are necessary to allow the reader to fully understand the work flow and (2) the results should be better interpreted and discussed in a physical manner to make them worth publishing. Below, I list some more specific comments and suggestions.

**SPECIFIC COMMENTS**

1. Section 2.1 needs some better motivation for some choices:
   a. Why has RCP 8.5 been chosen? This needs some motivation.
   b. Why is only data from 1906–2000 used for the historical period?
   c. Why is only one ensemble member per model used (r1i1p1)? I think the analysis might be more robust when an ensemble of model outputs is used.

2. Section 2.2 needs to be improved to fully understand the workflow:
   a. It needs to be clear from this section how the authors will deal with the model output from the models listed in Table S1. Will the authors average everything out or separately perform the analysis at every single model and compare the results to each other? Now, this is only clear from the figure captions.
   b. How do the authors deal with different spatial resolutions of the model outputs?
   c. At Line 78, the authors mention the temporal resolution of the analysis; but it is not clear when and why both annual and monthly aggregations are used. In addition, differences in the results from these two experiments are not properly addressed in the paper.
   d. Line 80: how is the optimal order of the regression model determined? Is this order different for every grid cell or the same across the globe?
   e. Line 86: how are the data normalized and de-trended? Why are the data de-trended?
   f. Given the importance of the Granger causality framework for this work, I think it is necessary to at least summarise it in this section. At this point, the reader is simply directed to literature.
   g. It has been shown that modes of internal climate variability might be significantly correlated with each other and that this correlation needs to be taken into account to properly analyse their effect on other variables (see e.g. Martens *et al.* (2018) or Gonsamo *et al.* (2016)). Also IOD and ENSO are correlated (see e.g. Figure S17 in Martens *et al.* (2018)). It is not clear to me how this is achieved by using the model described in Equation 1.
   h. How did the authors check the validity of Equation 1? Are the fitted models tested for significance?

3. As the authors correctly point out near the end of Section 2.1, several issues arise when using output from climate models. Both the modelled evaporation and the calculated climate indices are uncertain, and it is unclear to which extent this affects the analysis in the paper. I understand that the authors somehow try to tackle this by relying on the output from different models; but I think too little attention is given to this issue in the paper. I would at least expect a brief discussion about the possible uncertainties in the analysis: how reliable are the derived climate indices used to describe the IOD, ENSO, and NAO? The authors could for instance benchmark them against observed indices. How reliable is the evaporation in the models? Again, this can be done by benchmarking against in situ observations. Alternatively, the authors could discuss the uncertainties based on existing literature to put their results in context:

e.g. in which regions are the results presumably less reliable due to uncertainties in evaporation or internal climate variability?

4. The impact of IOD on evaporation over land is surprisingly very low; although it has been shown in several publications that the IOD is significantly affecting the surface hydrology; e.g. in Australia. How do the authors explain this low impact found in their study?

5. One of the main advantages of using output from climate models is the availability of surface and atmospheric variables driving evaporation, all linked by the model in a physical manner. As such, the observed patterns described in Section 3 can be better explained from a physical point of view in my opinion. Why are certain links between evaporation and the climate modes found (or not found) in specific regions? Most of the discussion is relatively speculative at the moment, while I think it should be feasible to explain the observed patterns by some additional analyses. Speculative sentences like "… *the influence of ENSO on evaporation might be associated with Wind-Evaporation SST*" (P4-L108), "*In the Northern Hemisphere, this result might be due to decrease in solar radiation.*" (P6-L165), "*This increase in ENSO impacts might be related to the increase …*" (P6-L177), or "*There are different factors that might contribute to the ambiguity of climate mode impacts on evaporation …*" (P7-L204) could be better answered, by also analysing the effect of the modes on other model variables.

6. The statements at P8-L237-238 and P9-L258-260 are confusing. Modes of climate variability affect surface meteorological variables that drive the evaporation process like precipitation, wind, and air temperature, which, in turn, affect evaporation. The fact that no clear link can be found between evaporation dynamics and the modes of climate variability does not necessarily mean that these drivers are more important to explain variability in evaporation, but rather indicates that the drivers are not affected by the modes of climate variability in the models.

7. I am a bit surprised that there is generally little difference between the results for the future and historical periods. Several studies have shown that the modes of climate variability analysed in this paper are affected by climate change, and that (e.g.) more extreme states of these modes are expected (this is also acknowledged in the paper several times). How do the authors explain this small difference?

**TECHNICAL CORRECTIONS**

1. P1-L28-29: "… *and are likely to have impacts on global evaporation and transpiration …*": it should be explained why this is expected, or the statement should be backed-up with references.

2. P2-L31-32: It is unclear what is meant by this statement. I think "indicator" is simply the wrong choice of word here; else, the authors need to add which aspect of e.g. the global water cycle is "indicated" by evaporation.

3. P2-L40: References should be given here to make clear about which "previous works" the authors are talking.

4. P2-L36-48: Please note that Martens *et al.* (2018) preformed a comprehensive analysis of the impact of 16 major modes (including the ones tested here) of climate variability on terrestrial evaporation. Although the paper is cited in the results section, I think it is fair to cite it here as well.

5. P3-L61-62: The importance of this statement for the paper is not clear.

6. P4-L98: Indian Oceans → Indian Ocean

7. Why do the authors use the probability of the absence of Granger causality, rather than the presence? To me this is rather confusing, especially when looking at figures. Also the discussion of Figure 5 at page 7 is complicated by this, I think.

8. Regarding Figures 1–4 and Figure 6:
   a. I would like to advice to use a different color map. The use of a "rainbow" color map is misleading and should be avoided (I encourage the authors to google this and find out the reasons).
   b. The labels indicating 60 and 90 degrees latitude (both south and north) overlap with the map.
   c. For the contours, I would use a color not used in the color map.
   d. The symbols used to indicate the lines of equal latitude and longitude should be different from the dot used to indicate the agreement between models. I would simply not plot the parallels and meridians to make the figures less busy.

9. Regarding Figure 5: this figure is currently useless in providing information on differences between IOD and NAO. Would it be possible to plot these on the right y-axis with a different scale?

**REFERENCES**

Gonsamo, A., Chen, J. M., Lombardozzi, D. Global vegetation productivity response to climatic oscillations during the satellite era. Global Change Biology 22 (10), 3414–3426 (2016).

Martens, B., Waegeman, W., Dorigo, W., Verhoest, N., Miralles, D. Terrestrial evaporation response to modes of climate variability. npj Climate and Atmospheric Science 1 (1), 43 (2018).

*Brecht Martens*
*Ghent University*
*Laboratory of Hydrology and Water Management*

---

## Author Comment (AC1) · 18 Nov 2019

**1 Response to Reviewer #1's comments**

HESS-2019-442

Le et al. (2019): Response of global evaporation to major climate modes in historical and future CMIP5 simulations

**GENERAL COMMENTS**

This paper explores the historical and future impact of three major modes of internal climate variability on evaporation from oceans and land into the atmosphere based on data from CMIP5 model simulations and a Granger causality framework. Such an analysis might provide useful insights about the distribution of water resources in the near future and to help better forecast extreme hydrological events. As such, I truly see scientific value in this study; however, in my opinion, the present paper first needs to be improved in two ways: (1) the description of the method needs to be improved and more details are necessary to allow the reader to fully understand the work flow and (2) the results should be better interpreted and discussed in a physical manner to make them worth publishing. Below, I list some more specific comments and suggestions.

**Response:** We appreciate the valuable comments and detailed suggestions from reviewer Brecht Martens. We agree that the description of the methods is not completely clear, and we provided additional information for this section. We also provided further discussions for the results as suggested by the reviewer to improve the manuscript. The responses to the reviewer's comments are provided below in **blue text**. The reviewer's comments are shown in **black text**.

**SPECIFIC COMMENTS**

*1. Section 2.1 needs some better motivation for some choices:*

a. Why has RCP 8.5 been chosen? This needs some motivation.

**Response:** We added the following text to Section 2.1 to clarify this motivation:

"The RCP8.5 is a very high emission scenario with radiative forcing of 8.5 W/m$^2$ in 2100 relative to the preindustrial level (van Vuuren et al., 2011). Warming environment in the RCP8.5

scenario increases the frequency of extreme ENSO and IOD events (Cai et al., 2014a, 2015) and potentially modulates the impacts of these climate modes on global evaporation.”

b. Why is only data from 1906–2000 used for the historical period?

**Response:** The original historical period in model simulations is 1850-2005. The original future period is 2006-2100. However, as we will compare the impacts of climate modes on evaporation between the historical period and future period, it is important to use the same data length for both periods. Thus, only data from 1906–2000 is used for the historical period. We modified the relative sentences in Section 2.1 to clarify this point:

“The results of the effects of climate modes on evaporation are compared between the historical period and the future period. Hence, in our analyses, we only use the data for the 1906-2000

historical period as a reference (with similar data length) for the future period 2006-2100”

c. Why is only one ensemble member per model used (r1i1p1)? I think the analysis might be more robust when an ensemble of model outputs is used.

**Response:** The total ensemble members are different between models (e.g., several models may provide up to ten ensemble members and others provide less), thus, it is challenging to determine the number of ensemble members (for each model) for analyses. Using one ensemble member per model is a simple way to guarantee the “one model, one vote” rule (Knutti et al., 2010). Here we use 15 different models for the analysis, this common approach is widely used and helps to reduce the uncertainties. We added the following text to Section 2.1 to clarify this point:

“As we use 15 different models for our analysis, the uncertainties related to the effects of climate modes on evaporation are reduced. The results based on multi-model mean were shown to be better and more reliable than single model results (Weigel et al., 2010).”

*2. Section 2.2 needs to be improved to fully understand the workflow:*

a. It needs to be clear from this section how the authors will deal with the model output from the models listed in Table S1. Will the authors average everything out or separately perform the analysis at every single model and compare the results to each other? Now, this is only clear from the figure captions.

**Response:** We added the following text to Section 2.2 to clarify this point:

"We apply the methods described above to all the single models. We then rescale the results of
single model to 1° longitude ×1° latitude spatial resolution. We use the rescaled results to
compute the multi-model mean which is shown as a map of probability for no Granger causal
impact from individual climate mode to global evaporation."

b. How do the authors deal with different spatial resolutions of the model outputs?
**Response:** We added the following text to Section 2.2 to clarify this point:
"We apply the methods described above to all the single models. We then rescale the results of
single model to 1° longitude ×1° latitude spatial resolution. We use the rescaled results to
compute the multi-model mean which is shown as a map of probability for no Granger causal
impact from individual climate mode to global evaporation."

c. At Line 78, the authors mention the temporal resolution of the analysis; but it is not clear when
and why both annual and monthly aggregations are used. In addition, differences in the results
from these two experiments are not properly addressed in the paper.
**Response:** We thank the reviewer for pointing this out. Here the temporal resolution of all
analyses is yearly. We only change the definition of the predictand (i.e., $X_t$) from annual mean to
seasonal mean. This change in definition does not alter the temporal resolution of the predictand
(i.e. the temporal resolution is yearly for both definitions used). Regarding the difference
between these two experiments, we consider the analyses using the annual mean of evaporation
(i.e. the predictand $X_t$) are the main results (Figures 1, 2, 3) while the analyses using the seasonal
mean of evaporation provide additional information (Figures S3 to S10). We add the following
text to Section 2.2 to clarify this point:
"We note that the temporal resolution of all analyses is yearly. Although the definition of the
predictand (i.e., $X_t$ in equation (1)) is based on both annual mean and seasonal mean values, the
change in definition does not alter the temporal resolution of the predictand (i.e., the temporal
resolution is yearly for both definitions used). We report the analyses using the annual mean of
evaporation (i.e., the predictand $X_t$) as the main results of this study while the analyses using the
seasonal mean of evaporation provide additional information."

d. Line 80: how is the optimal order of the regression model determined? Is this order different
for every grid cell or the same across the globe?

**Response:** We added the following text to Section 2.2 to clarify these points:

"The optimal order p is computed by minimizing the Bayesian information criterion or Schwarz criterion (Schwarz, 1978). We note that the optimal orders might be different for each grid cell, depending on evaporation data of the selected grid cell."

e. Line 86: how are the data normalized and de-trended? Why are the data de-trended?

**Response:** The purpose of normalizing and detrending is only to simplify the data time series without altering the results and conclusions. We modified the related sentence in Section 2.2 to clarify this point as below:

"All climate indices and evaporation data are normalized (by using z-score) and detrended (by subtracting the trending line from given data; the trending line or the best-fit line is identified using least squares method). Detrending the data does not alter the results and conclusions."

f. Given the importance of the Granger causality framework for this work, I think it is necessary to at least summarise it in this section. At this point, the reader is simply directed to literature.

**Response:** We added the summary of Granger causality test in Section 2.2 as below:

"The model shown in equation (1) is defined as a complete predictive model where all variables (i.e., past data of evaporation and climate indices) are used to estimate evaporation. The null model of no causal effects from given climate mode (i.e., variable $Y$) to evaporation is defined by removing the terms related to $Y$ (i.e., by setting $\beta_i = 0 \ with \ i = 1, \dots, p$) in equation (1). The complete model and the null model are then compared by using the following indicator:

$$L_{Y \to X} = n(\log\left|\Omega_{p,\beta_i=0}\right| - \log\left|\Omega_p\right|)$$

where $|\Omega_p|$ is the determinant of the covariance matrix of the noise residual, and n is the length of the data time series. We test the significance of the complete model by comparing the $L_{Y \to X}$

indicator against a $\chi_p^2$ null distribution. This test results in a probability for no causal effect of the considered variable $Y$ on evaporation."

g. It has been shown that modes of internal climate variability might be significantly correlated with each other and that this correlation needs to be taken into account to properly analyse their effect on other variables (see e.g. Martens et al. (2018) or Gonsamo et al. (2016)). Also IOD and

ENSO are correlated (see e.g. Figure S17 in Martens et al. (2018)). It is not clear to me how this is achieved by using the model described in Equation 1.

**Response:** We think the methods used in our study are not directly related to correlation analyses. More importantly, the result of Granger causality test is independent from the relationship of predictors (e.g., ENSO and the IOD) (Mosedale et al., 2006; Stern and Kaufmann,

2013). In fact, with the approach described above and below (see the responses to comments 2f and 2h), the methods used account for the characteristics of all climate indices, including the effect of cross-correlated between IOD and ENSO. Specifically, the complete predictive model shown in Equation 1 partly accounts for possible correlation between ENSO and the IOD by automatically adjusting the regression coefficients $\alpha_i$, $\beta_i$ and $\delta_{j,i}$ and the noise residuals $\varepsilon_t$, based on the characteristics of ENSO and the IOD.

We added the following sentences to Section 2.2 to clarify this point:

"Modes of climate variability might be correlated to each other and this correlation might have effects on the relationship between these modes and other variables (e.g., evaporation) (Gonsamo et al., 2016; Martens et al., 2018). However, in the approach of Granger causal analysis, the conclusion for the causal effects from variable Y (i.e., the considered climate mode) to variable

X (i.e., evaporation) is independent from the relationship between Y and other factors (i.e., the relationship between climate modes) (Mosedale et al., 2006; Stern and Kaufmann, 2013)."

h. How did the authors check the validity of Equation 1? Are the fitted models tested for significance?

**Response:** The model shown in Equation 1 is defined as a complete model where all variables (i.e., past data of evaporation and climate indices) are used to predict evaporation. We evaluate the validity of the complete model by comparing this model with a null model. The null model of no causal effects from given climate mode (i.e. variable Y) to evaporation is defined by removing the terms related to Y (i.e., by setting $\beta_i = 0 \ with \ i = 1, ..., p$) in Equation 1. The complete model and the null model are then compared by using the following indicator:

$L_{Y \rightarrow X} = n(\log|\Omega_{p,\beta_i=0}| - \log|\Omega_p|)$

where $|\Omega_p|$ is the determinant of the covariance matrix of the noise residual, and n is the length of the data time series. We test the significance of the full model by comparing the $L_{Y \rightarrow X}$ indicator against a $\chi_p^2$ null distribution. This test results in a probability for no causal effect of the considered variable Y on evaporation.

We added the following text to Section 2.2 to clarify these points:

"The model shown in equation (1) is defined as a complete predictive model where all variables (i.e., past data of evaporation and climate indices) are used to estimate evaporation. The null model of no causal effects from given climate mode (i.e., variable $Y$) to evaporation is defined by removing the terms related to $Y$ (i.e., by setting $\beta_i = 0\ with\ i = 1, ..., p$) in equation (1). The complete model and the null model are then compared by using the following indicator:

$$L_{Y \to X} = n(\log\left|\Omega_{p,\beta_i=0}\right| - \log\left|\Omega_p\right|)$$

where $|\Omega_p|$ is the determinant of the covariance matrix of the noise residual, and n is the length of the data time series. We test the significance of the complete model by comparing the $L_{Y \to X}$

indicator against a $\chi_p^2$ null distribution. This test results in a probability for no causal effect of the considered variable $Y$ on evaporation."

*3. As the authors correctly point out near the end of Section 2.1, several issues arise when using*

*output from climate models. Both the modelled evaporation and the calculated climate indices*

*are uncertain, and it is unclear to which extent this affects the analysis in the paper. I understand*

*that the authors somehow try to tackle this by relying on the output from different models; but I*

*think too little attention is given to this issue in the paper. I would at least expect a brief*

*discussion about the possible uncertainties in the analysis: how reliable are the derived climate*

*indices used to describe the IOD, ENSO, and NAO? The authors could for instance benchmark*

*them against observed indices. How reliable is the evaporation in the models? Again, this can be*

*done by benchmarking against in situ observations. Alternatively, the authors could discuss the*

*uncertainties based on existing literature to put their results in context: e.g. in which regions are*

*the results presumably less reliable due to uncertainties in evaporation or internal climate*

*variability?*

**Response:** We thank the reviewer for these suggestions. The uncertainties related to the simulations of climate indices and evaporation are discussed in previous works and we cited as below:

"There might exist model biases in simulating ENSO (e.g. Taschetto et al., 2014), the IOD (Chu et al., 2014; Weller and Cai, 2013), NAO (Gong et al., 2017; Lee et al., 2018) and there is uncertainty in capability of land surface models in modeling evaporation (Mueller &

Seneviratne, 2014; Wang & Dickinson, 2012)."

Despite these uncertainties, the CMIP5 simulations are still very helpful and these datasets are widely used (e.g., Cai et al., 2014b, 2015). As noted by the reviewer the approach of multi-model mean partly address the issue of uncertainties related to both evaporation and internal climate variability as simulated by climate models. The results described in our study also consider these uncertainties (i.e., by using significance level and agreement level between models). We noted that the high uncertainties for the ENSO effects are only shown for several regions (e.g. South

Asia, Africa and Southern South America).

The following sentences in Section 3 (Discussions) discuss the uncertainties in simulating climate modes and evaporation:

"There are different factors that might contribute to the ambiguity of climate mode impacts on evaporation of several regions (e.g. South Asia, Africa and Southern South America).

Specifically, these factors include the large discrepancies of current estimations of land evaporation for recent decades (Dong & Dai, 2017; Miralles et al., 2016), the limitations of climate models in simulating climate modes (Gong et al., 2017; Lee et al., 2018; Taschetto et al.,

2014; Weller and Cai, 2013) and the overestimation of simulated evaporation in most regions (Mueller and Seneviratne, 2014)."

We also added the following sentences to Section 3 (Discussions) to further clarify the uncertainties related to model simulations:

"Specifically, there are systematic biases in simulating yearly average evaporation in Australia,

China, Western North America, Europe, Africa and part of Amazonia (Mueller and Seneviratne,

2014). Thus, these biases contribute to the uncertainties in the effects of climate modes on evaporation. Nevertheless, the methods based on multi-model mean and Granger causality tests (see Section 2.2) help to reduce the uncertainties and provide robust results and conclusions."

*4. The impact of IOD on evaporation over land is surprisingly very low; although it has been*

*shown in several publications that the IOD is significantly affecting the surface hydrology; e.g.*

*in Australia. How do the authors explain this low impact found in their study?*

**Response:** We think these differences might come from different approaches in previous works and our study. For example, the impacts of the IOD might not be assessed with the contributions of confounding factors (e.g., ENSO and NAO). We think for several specific IOD events (e.g., extreme IOD or the IOD events associated with weak ENSO events), the IOD may significantly influence evaporation and surface hydrology in Australia. However, the results of our study imply that the multi-year impacts of the IOD on evaporation of Australia is not significant. Thus, we think this is not completely a contradiction.

*5. One of the main advantages of using output from climate models is the availability of surface and atmospheric variables driving evaporation, all linked by the model in a physical manner. As such, the observed patterns described in Section 3 can be better explained from a physical point of view in my opinion. Why are certain links between evaporation and the climate modes found (or not found) in specific regions? Most of the discussion is relatively speculative at the moment, while I think it should be feasible to explain the observed patterns by some additional analyses. Speculative sentences like "... the influence of ENSO on evaporation might be associated with Wind-Evaporation SST" (P4-L108), "In the Northern Hemisphere, this result might be due to decrease in solar radiation. " (P6-L165), "This increase in ENSO impacts might be related to the increase ..." (P6-L177), or "There are different factors that might contribute to the ambiguity of climate mode impacts on evaporation ..." (P7-L204) could be better answered, by also analysing the effect of the modes on other model variables.*

**Response:** We thank the reviewer for these suggestions.

- Regarding the sentence "… the influence of ENSO on evaporation might be associated with Wind-Evaporation SST" (P4-L108), we added supporting Figure S2 and rewrote this sentence as below:

"Further analyses reveal that ENSO has significant impacts on SST (Figure S2a) and zonal winds (Figure S2b) over the tropical Pacific for the 1906-2000 period (similar patterns are observed for the 2006-2100 period, not shown). Hence, the influence of ENSO on evaporation might be associated with Wind-Evaporation SST (WES) effect (Cai et al., 2019). The WES effect occurs when warm (cold) water becomes warmer (colder) due to decrease (increase) in evaporation and weakened (strengthened) surface winds."

We also added the following sentence to the Section "ENSO influence on evaporation":

"ENSO causal effects on global precipitation are shown in Figure S2c which indicates the close connection between precipitation and evaporation process in several regions (e.g., tropical

Pacific, Australia, Amazonia and regions closed to Caspian Sea)."

a)

[Figure]

b)

[Figure]

c)

[Figure]

MODELS MEAN: ENSO - PRECIPITATION PERIOD 1906-2000

**Figure S2.** Multi-model mean probability map for the absence of Granger causality between ENSO and
annual mean SST (a), zonal winds (b) and precipitation (c) for the period 1906-2000. Stippling
demonstrates that at least 70% of models show agreement on the mean probability of all models at given
grid point. An individual model's agreement is determined when the difference between the multi-model
mean probability and the selected model's probability is less than one standard deviation of multi-model
mean probability. The green (red) contour line designates $p$ value = 0.1 (0.05). Brown shades indicate low
probability for the absence of Granger causality. ENSO = El Niño–Southern Oscillation. SST = sea
surface temperature.

- Regarding the sentence "In the Northern Hemisphere, this result might be due to decrease in solar radiation. " (P6-L165)", we added a reference as below:

"In the Northern Hemisphere, this result might be due to decrease in seasonal solar radiation (Martens et al., 2018)."

- Regarding the sentence "This increase in ENSO impacts might be related to the increase …"

(P6-L177), we removed this sentence to avoid confusing the readers.

- Regarding the sentence "There are different factors that might contribute to the ambiguity of climate mode impacts on evaporation …" (P7-L204), we removed the word "might" in this sentence. We noted that in the next sentence, we provided some references for clarification.

Overall, we think the effects of climate modes on a single driver of evaporation do not imply the effects of climate modes on evaporation. This result points to the complexity of evaporation processes which are influenced by different factors.

*6. The statements at P8-L237-238 and P9-L258-260 are confusing. Modes of climate variability affect surface meteorological variables that drive the evaporation process like precipitation, wind, and air temperature, which, in turn, affect evaporation. The fact that no clear link can be found between evaporation dynamics and the modes of climate variability does not necessarily mean that these drivers are more important to explain variability in evaporation, but rather indicates that the drivers are not affected by the modes of climate variability in the models.*

**Response:** We thank the reviewer for raising this point. We re-structure these statements as below:

"These results suggest that, for several regions of declining impacts of climate modes (highlighted in blue shades, Figure 6), the important drivers of evaporation processes in the 21$^{st}$ century (e.g., precipitation, near-surface air temperature, wind speed, soil moisture) tend to be not affected by the modes of climate variability in the models."

"Land evaporation is shown to have weak connection with teleconnection indices in several regions, suggesting the weak effects of climate modes on important drivers of land evaporation, such as local wind speed (Stephens et al., 2018), surface temperature (Laîné et al., 2014; Miralles et al., 2013), moisture supply (Jung et al., 2010) and amount of precipitation (Parr et al., 2016)."

*7. I am a bit surprised that there is generally little difference between the results for the future and historical periods. Several studies have shown that the modes of climate variability analysed in this paper are affected by climate change, and that (e.g.) more extreme states of these modes are expected (this is also acknowledged in the paper several times). How do the authors explain this small difference?*

**Response:** We think this small difference is due to the nature of methods used in this study. Here we assess the multi-year effects of climate modes on evaporation rather than the effects of single event. Thus, the effects on evaporation of the extreme states of these modes do not persist long enough to be significant. Moreover, in the climate system, the effects of these extreme IOD events might be compensated by the extreme events of other climate modes (e.g., ENSO).

We added the following sentence to Section 4 to clarify this point:

"These results imply that the effects on evaporation of the extreme states of the IOD do not persist long enough to be significant. Moreover, in the climate system, the effects of these extreme IOD events might be compensated by the extreme events of other climate modes (e.g., ENSO)."

**TECHNICAL CORRECTIONS**

*1. P1-L28-29: "… and are likely to have impacts on global evaporation and transpiration …":* *it should be explained why this is expected, or the statement should be backed-up with* *references.*

**Response:** We thank the reviewer for this suggestion. We rewrote these sentences as below to clarify this point:

"These climate modes may have influence on important drivers of evaporation such as surface temperature (e.g., Arora et al., 2016; Leung & Zhou, 2016; Sun et al., 2016; Thirumalai et al., 2017; Wang et al., 2017), precipitation (Dai and Wigley, 2000), soil moisture (Nicolai-Shaw et al., 2016), humidity (Hegerl et al., 2015) and wind speed (Hurrell et al., 2003; Yeh et al., 2018). Hence, these climate modes are likely to have impacts on global evaporation and transpiration (hereafter simply referred as 'evaporation')."

*2. P2-L31-32: It is unclear what is meant by this statement. I think "indicator" is simply the* *wrong choice of word here; else, the authors need to add which aspect of e.g. the global water* *cycle is "indicated" by evaporation.*

**Response:** We modified "indicator" to "variable contributing to"

*3. P2-L40: References should be given here to make clear about which "previous works" the* *authors are talking.*

**Response:** We corrected this sentence as follows:

"Moreover, most of previous works (e.g., Shinoda and Han, 2005; Xing et al., 2016; Zveryaev and Hannachi, 2011) mainly address the connection between individual climate mode and evaporation, however, the role of other climate modes might not be included in the analyses."

*4. P2-L36-48: Please note that Martens et al. (2018) preformed a comprehensive analysis of the*
*impact of 16 major modes (including the ones tested here) of climate variability on terrestrial*
*evaporation. Although the paper is cited in the results section, I think it is fair to cite it here as*
*well.*

**Response:** We thank the reviewer for this suggestion. We rewrote the sentences in the
Introduction as below to include the results shown in the study of Martens et al. (2018):
"While previous studies emphasize the importance of ENSO (Martens et al., 2018; Miralles et
al., 2013), the Atlantic Multidecadal Oscillation, the Tropical Northern Atlantic Dipole, Tropical
Southern Atlantic Dipole and the IOD (Martens et al., 2018) on global land evaporation, the role
of the NAO remains elusive."

*5. P3-L61-62: The importance of this statement for the paper is not clear.*
**Response:** We added the following sentence to this statement to clarify its purpose:
"Hence, the term 'evaporation' used in this study is referred to both transpiration and
evaporation."

*6. P4-L98: Indian Oceans → Indian Ocean*
**Response:** We modified the text

*7. Why do the authors use the probability of the absence of Granger causality, rather than the*
*presence? To me this is rather confusing, especially when looking at figures. Also the discussion*
*of Figure 5 at page 7 is complicated by this, I think.*
**Response:** We use the probability of the absence of Granger causality because the test of
Granger causality is based on the null hypothesis of no Granger causal effects from climate
modes to evaporation (see also the responses in section 2f and 2h of SPECIFIC COMMENTS).
The probability is computed and shown for no causal effects. The presence of Granger causality
is not directly tested, but it is an inference. We rewrote the first sentence of this paragraph as
below to clarify this point:
"Figure 5 shows the fraction area of Earth surface for land and ocean with probability for the
absence of Granger causality between climate modes and evaporation less than 0.1 (i.e., p value
< 0.1; here, the null hypothesis of no Granger causality from climate modes to evaporation is rejected at 10% significance level, hence, we conclude that there is significant causal effects; we note that the fraction area is substantially smaller if p value < 0.05).”

*8. Regarding Figures 1–4 and Figure 6:*

a. I would like to advice to use a different color map. The use of a "rainbow" color map is misleading and should be avoided (I encourage the authors to google this and find out the reasons).

**Response:** We thank the reviewer for this advice. We changed the color map of Figures 1-3 to a more perceptually-uniform one (brown-to-white color scale).

Regarding Figures 4 and 6, we changed the color map to blue-white-brown color scale. The map represents both negative and positive p-value.

We showed here Figures 1 and 4.

[Figure]

**Figure 1.** Multi-model mean probability map for the absence of Granger causality between ENSO and annual mean evaporation for the periods 1906-2000 (a) and 2006-2100 (b). Stippling demonstrates that at least 70% of models show agreement on the mean probability of all models at given grid point. An individual model's agreement is determined when the difference between the multi-model mean probability and the selected model's probability is less than one standard deviation of multi-model mean probability. The green (red) contour line designates $p$ value = 0.1 (0.05). Brown shades indicate low probability for the absence of Granger causality. ENSO = El Niño–Southern Oscillation.

[Figure]

**Figure 4.** Difference in multi-model mean probability for the absence of Granger causality between a pair of climate modes and annual mean evaporation. The results are shown for the periods 1906-2000 (a, b, c) and 2006-2100 (d, e, f). ENSO minus NAO (a, d). ENSO minus IOD (b, e). NAO minus IOD (c, f). Blue shades indicate lower probability for the absence of Granger causality. ENSO = El Niño–Southern Oscillation. NAO = North Atlantic Oscillation. IOD = Indian Ocean Dipole.

b. The labels indicating 60 and 90 degrees latitude (both south and north) overlap with the map.

**Response:** We thank the reviewer for this suggestion. We removed all the longitude and latitude labels in the figures.

c. For the contours, I would use a color not used in the color map.

**Response:** We thank the reviewer for this suggestion. We changed the color of the contours.

d. The symbols used to indicate the lines of equal latitude and longitude should be different from the dot used to indicate the agreement between models. I would simply not plot the parallels and meridians to make the figures less busy.

**Response:** We thank the reviewer for this advice. We removed the latitudinal and longitudinal grids to make the figures easier to read.

*9. Regarding Figure 5: this figure is currently useless in providing information on differences*

*between IOD and NAO. Would it be possible to plot these on the right y-axis with a different*

*scale?*

**Response:** We agree with the reviewer that it is difficult to distinguish the difference between the IOD and NAO. This is because the fraction area influenced by these 2 modes is very small and close to zero. In fact, the fraction areas of NAO are zeros in Figures 5a, c and d. The fraction area for the effects of the IOD and NAO is much smaller compared to the fraction area of ENSO.

Thus, it is difficult to plot all these information. We added the following text in the Figure 5

caption to clarify this point:

"Several fraction areas are close to zero"

We should note that we have additional Figure S11 with similar information, but the fraction areas are computed for p value < 0.25 (i.e. climate modes are unlikely to have no causal effects on evaporation). This Figure might be used as an alternative to compare the difference between the IOD and NAO. We added the following text to the Discussions Section:

"Figure S11 indicates that the land and ocean area influenced by the IOD is slightly higher compared to NAO."

**REFERENCES**

Gonsamo, A., Chen, J. M., Lombardozzi, D. Global vegetation productivity response to climatic oscillations during the satellite era. Global Change Biology 22 (10), 3414–3426 (2016).

Martens, B., Waegeman, W., Dorigo, W., Verhoest, N., Miralles, D. Terrestrial evaporation response to modes of climate variability. npj Climate and Atmospheric Science 1 (1), 43 (2018).

*Brecht Martens*

*Ghent University*

*Laboratory of Hydrology and Water Management*

**2 References**

Cai, W., Santoso, A., Wang, G., Weller, E., Wu, L., Ashok, K., Masumoto, Y. and Yamagata, T.: Increased frequency of extreme Indian Ocean Dipole events due to greenhouse warming., Nature, 510(7504), 254–8, doi:10.1038/nature13327, 2014a.

Cai, W., Borlace, S., Lengaigne, M., van Rensch, P., Collins, M., Vecchi, G., Timmermann, A., Santoso, A., McPhaden, M. J., Wu, L., England, M. H., Wang, G., Guilyardi, E. and Jin, F.-F.: Increasing frequency of extreme El Niño events due to greenhouse warming, Nat. Clim. Chang., 5(2), 1–6, doi:10.1038/nclimate2100, 2014b.

Cai, W., Wang, G., Santoso, A., McPhaden, M. J., Wu, L., Jin, F.-F., Timmermann, A., Collins, M., Vecchi, G., Lengaigne, M., England, M. H., Dommenget, D., Takahashi, K. and Guilyardi, E.: Increased frequency of extreme La Niña events under greenhouse warming, Nat. Clim. Chang., 5(2), 132–137, doi:10.1038/nclimate2492, 2015.

Dong, B. and Dai, A.: The uncertainties and causes of the recent changes in global evapotranspiration from 1982 to 2010, Clim. Dyn., 49(1–2), 279–296, doi:10.1007/s00382-016-3342-x, 2017.

Gong, H., Wang, L., Chen, W., Chen, X. and Nath, D.: Biases of the wintertime Arctic Oscillation in CMIP5 models, Environ. Res. Lett., 12(1), doi:10.1088/1748-9326/12/1/014001, 2017.

Knutti, R., Furrer, R., Tebaldi, C., Cermak, J. and Meehl, G. a.: Challenges in Combining Projections from Multiple Climate Models, J. Clim., 23(10), 2739–2758, doi:10.1175/2009JCLI3361.1, 2010.

Lee, J., Sperber, K. R., Gleckler, P. J., Bonfils, C. J. W. and Taylor, K. E.: Quantifying the agreement between observed and simulated extratropical modes of interannual variability, Springer Berlin Heidelberg., 2018.

Martens, B., Waegeman, W., Dorigo, W. A., Verhoest, N. E. C. and Miralles, D. G.: Terrestrial evaporation response to modes of climate variability, npj Clim. Atmos. Sci., 1(1), 43, doi:10.1038/s41612-018-0053-5, 2018.

Miralles, D. G., van den Berg, M. J., Gash, J. H., Parinussa, R. M., de Jeu, R. a. M., Beck, H. E., Holmes, T. R. H., Jiménez, C., Verhoest, N. E. C., Dorigo, W. a., Teuling, A. J. and Johannes

Dolman, A.: El Niño–La Niña cycle and recent trends in continental evaporation, Nat. Clim.
Chang., 4(1), 1–5, doi:10.1038/nclimate2068, 2013.

Miralles, D. G., Jiménez, C., Jung, M., Michel, D., Ershadi, A., Mccabe, M. F., Hirschi, M.,
Martens, B., Dolman, A. J., Fisher, J. B., Mu, Q., Seneviratne, S. I., Wood, E. F. and Fernández-
Prieto, D.: The WACMOS-ET project - Part 2: Evaluation of global terrestrial evaporation data
sets, Hydrol. Earth Syst. Sci., 20(2), 823–842, doi:10.5194/hess-20-823-2016, 2016.

Mueller, B. and Seneviratne, S. I.: Systematic land climate and evapotranspiration biases in
CMIP5 simulations, Geophys. Res. Lett., 41(1), 128–134, doi:10.1002/2013GL058055, 2014.

Shinoda, T. and Han, W.: Influence of the Indian Ocean dipole on atmospheric subseasonal
variability, J. Clim., 18(18), 3891–3909, doi:10.1175/JCLI3510.1, 2005.

Taschetto, A. S., Gupta, A. Sen, Jourdain, N. C., Santoso, A., Ummenhofer, C. C. and England,
M. H.: Cold tongue and warm pool ENSO Events in CMIP5: Mean state and future projections,
J. Clim., 27(8), 2861–2885, doi:10.1175/JCLI-D-13-00437.1, 2014.

van Vuuren, D. P., Edmonds, J., Kainuma, M., Riahi, K., Thomson, A., Hibbard, K., Hurtt, G.
C., Kram, T., Krey, V., Lamarque, J. F., Masui, T., Meinshausen, M., Nakicenovic, N., Smith, S.
J. and Rose, S. K.: The representative concentration pathways: An overview, Clim. Change,
109(1), 5–31, doi:10.1007/s10584-011-0148-z, 2011.

Weigel, A. P., Knutti, R., Liniger, M. a. and Appenzeller, C.: Risks of Model Weighting in
Multimodel Climate Projections, J. Clim., 23(15), 4175–4191, doi:10.1175/2010JCLI3594.1,
2010.

Weller, E. and Cai, W.: Realism of the indian ocean dipole in CMIP5 models: The implications
for climate projections, J. Clim., 26(17), 6649–6659, doi:10.1175/JCLI-D-12-00807.1, 2013.

Xing, W., Wang, W., Shao, Q., Yu, Z., Yang, T. and Fu, J.: Periodic fluctuation of reference
evapotranspiration during the past five decades: Does Evaporation Paradox really exist in
China?, Sci. Rep., 6(August), 1–12, doi:10.1038/srep39503, 2016.

Zveryaev, I. I. and Hannachi, A. a.: Interannual variability of Mediterranean evaporation and its
relation to regional climate, Clim. Dyn., 38(3–4), 495–512, doi:10.1007/s00382-011-1218-7,
2011.

---

## Referee Comment (RC2) · Jasper Denissen (Referee) · 19 Nov 2019

**General comments**
The authors discuss the causal effects of climate modes on past and future (global) evaporation using an ensemble of simulations. With a (relatively) simple metric the authors show whether changes in (land or ocean) evaporation are likely to be caused by climate modes. The authors have done an impressive literature study, which backs up their own data-rich analysis. The main conclusion is basically that individual climate modes have effects all over the globe. The potential of this paper lies in using such a wealth of data (causal effects of different climate modes on global evaporation and

its consistency across an ensemble of climate model simulations). In the end, most of the results are presented separately (per climate mode). I would advise to synthesize all causal effects of different climate modes in one global figure, where the authors could show in which regions (past or future) which climate mode is dominant. The text abundantly mentions which areas are influenced most by which climate mode, but visualizing this in a figure would give a much better overview in my opinion.

**Specific comments**

- Line 43: "*Besides, evaluating the models' consistency in reproducing the impacts of internal climate variability on evaporation is important for understanding the difference between models*". I do think this is important, but hardly mentioned in the paper. This could be emphasized more (in the abstract/conclusions?).

- Line 54-55: Why are the starting/ending years 'roughly' and not exactly...?

- Line 59: From which climate model does 'rlilpl' come? Why only one member?

- Line 61-62: "*Most of climate models do not provide separately the data of evaporation from canopy (i.e. transpiration) and water evaporation from soil (i.e. evaporation).*" What does this mean for this research? I miss a connecting sentence here. Maybe append something like: "which complicates attributing changes in evaporation to canopy or soil related processes."

- Line 65: " ..., CMIP5 data is useful..." Why is the data useful? Because of the abundance of data based on climate models with slightly different assumptions?

- Line 73-74: Why focus on the tropical Pacific, tropical Indian and North Atlantic Oceans with the mentioned climate modes? Please motivate why these (and related climate modes) are in focus.

- Line 79: Please elaborate on what the order of the causal model means.

[Figure]

- Line 90: What is it that makes the applied techniques necessarily robust?

- Line 100-102: "*Specifically, high agreement of climate models in teleconnection between ENSO and tropical ocean evaporation implies that models can simulate the impacts of ENSO on evaporation.*" Why does a high agreement between models necessarily signal a capability to simulate impacts of ENSO on evaporation?

- Line 105: Why focus on two significance levels and not just on one? If two significance levels are necessary, please indicate why.

- Line 145: Figure S3 and S4 do not show seasonal responses of evaporation to the IOD. Do you mean Figure S5 and S6?

- Line 170: The differences in Fig. 4 do not imply unique combinations: a difference of -.1 can result from Granger causalities 0 and .1 or .9 and 1... I would advise to test whether the difference is significant or not.

- Line 185-194: Are these results or do these belong in the discussion section?

- Line 209: In Figure 5, panels a and b can be combined, just as panels c and d to be able to be able to visually compare a bit better. The authors could even think of combining all the panels together, to see be able to compare easily between land and ocean, past and future (there are only 12 bars).

- Line 217-220: What is the added value of Figure S10 (showing Figure 5 with p 0.25)? The conclusion remains the same and inferences of this figure aren't mentioned.

- Line 231: Same comment as for Figure 4: I would advise to test whether the means are significantly different or not.

- Line 231-232: "*Specifically, the fraction area of Earth surface showing lower probability of ENSO effects for 2006-2100 period is approximately 52.9% (Figure 6a).*" I highly doubt this conclusion. The authors should first assess whether the differences are significant or not and after that determine reassess the overall ENSO effects. 52.9% is just slightly higher than just assigning an de(in)crease of ENSO effects randomly (50%). This also goes for following similar conclusions.

**Technical corrections**

- Line 40: ". . . between individual climate mode**s**"

- Line 56: "Using other data periods with similar length**s** (i.e., 95 years) **do** not alter. . ."

- Line 61: "Most of **the** climate models. . ."

- Line 158: ". . .of NAO impact**s**. . ."

- Line 187: ". . .in agreement with previous studie**s**. . ."

- Line 198: "in **the** Australian continent. . ."

- Line 229-230: ". . .for **the** future period 2006-2100 and **the** historical period 1906-2000. . ."

- Line 264: ". . . for a short term. . ."

---

## Author Comment (AC2) · 8 Dec 2019

**1 Response to Reviewer #2's comments**

Jasper Denissen (Referee)

jdenis@bgc-jena.mpg.de

**General comments**

The authors discuss the causal effects of climate modes on past and future (global) evaporation using an ensemble of simulations. With a (relatively) simple metric the authors show whether changes in (land or ocean) evaporation are likely to be caused by climate modes. The authors have done an impressive literature study, which backs up their own data-rich analysis. The main conclusion is basically that individual climate modes have effects all over the globe. The potential of this paper lies in using such a wealth of data (causal effects of different climate modes on global evaporation and its consistency across an ensemble of climate model simulations). In the end, most of the results are presented separately (per climate mode). I would advise to synthesize all causal effects of different climate modes in one global figure, where the authors could show in which regions (past or future) which climate mode is dominant. The text abundantly mentions which areas are influenced most by which climate mode, but visualizing this in a figure would give a much better overview in my opinion.

**Response:** We thank Reviewer Jasper Denissen for helpful comments. In this document, the responses to the Reviewer's comments are provided below in **blue text**. The Reviewer's comments are shown in **black text**.

We added Figure 7 as a summary for the dominance of individual climate modes on regional evaporation as below. We chose to only show the regions with $p$ value less than 0.25, (i.e., climate modes are unlikely to have no causal effects on evaporation (Stocker et al., 2013)).

We also added the corresponding text to Discussions section:

25 "The dominance of an individual climate modes on evaporation is summarized in Figure 7 for
26 historical and future periods. Figure 7 shows the regions where the lowest probability for the
27 absence of Granger causality between climate modes and evaporation is less than 0.25 (i.e.,
28 climate modes are unlikely to have no causal effects on evaporation). This result indicates the
29 important role of ENSO on global evaporation."

MODELS MEAN OF PREDOMINANCE BETWEEN ENSO, NAO AND IOD - EVAPORATION: PERIOD 1906-2000

[Figure]

| ENSO | NAO | IOD |

MODELS MEAN OF PREDOMINANCE BETWEEN ENSO, NAO AND IOD - EVAPORATION: PERIOD 2006-2100

[Figure]

| ENSO | NAO | IOD |

**Figure 1.** The predominance of single climate mode on regional evaporation for periods 1906-2000 (a) and 2006-2100 (b). The predominance of a climate mode at a grid point is defined when the lowest $p$ value of all climate modes (see also Figures 1, 2 and 3) at the given grid point is less than 0.25 (i.e., climate modes are unlikely to have no causal effects on evaporation). The predominance of ENSO, NAO and the IOD on evaporation are shown in red, blue and green shades, respectively. ENSO = El Niño–Southern Oscillation. NAO = North Atlantic Oscillation. IOD = Indian Ocean Dipole.

**Specific comments**

• Line 43: "Besides, evaluating the models' consistency in reproducing the impacts of internal climate variability on evaporation is important for understanding the difference between models". I do think this is important, but hardly mentioned in the paper. This could be emphasized more (in the abstract/conclusions?).

**Response:** We thank the Reviewer for raising this point. We discussed the consistency of model simulations in Section 3. For example: "There is high agreement between models (indicated by stippling in Figure 1) in simulating ENSO-evaporation connection of these regions. Specifically, high agreement of climate models in teleconnection between ENSO and tropical oceans evaporation implies that models can simulate the impacts of ENSO on evaporation."

and

"In historical simulations (Figure 2a), the IOD impacts might reach as far as the Southern Ocean (region close to 150°W to 120°W; 45°S to 60°S) where there is high agreement between models."

and

"NAO mainly contributes to change in evaporation of the North Atlantic European sector where high agreement between models is found (see Figure S7 for additional details)."

We added the following sentence to the Abstract and Conclusions as your suggestion:

"There is high agreement between models in simulating the effects of climate modes on evaporation of these regions."

• Line 54-55: Why are the starting/ending years 'roughly' and not exactly...?

**Response:** Most models have exactly starting/ending years of 1850/2005 for historical simulation and starting/ending years of 2006/2100 for future simulation. Several models have different starting/ending years. For example, model MIROC5 has the ending year of 2012 in historical simulation. We think it is more correct to use the word 'roughly'.

• Line 59: From which climate model does 'rlilpl' come? Why only one member?

**Response:** All models have the simulation r1i1p1. Using one ensemble member per model is a simple way to guarantee the "one model, one vote" rule (Knutti et al., 2010). Here we use 15

different models for the analysis, this common approach is widely used and helps to reduce the uncertainties. We added the following text to Section 2.1 to clarify this point:

"As we use 15 different models for our analysis, the uncertainties related to the effects of climate modes on evaporation are reduced. The results based on multi-model mean were shown to be better and more reliable than single model results (Weigel et al., 2010)."

• Line 61-62: "Most of climate models do not provide separately the data of evaporation from canopy (i.e. transpiration) and water evaporation from soil (i.e. evaporation)." What does this mean for this research? I miss a connecting sentence here. Maybe append something like: "which complicates attributing changes in evaporation to canopy or soil related processes."

**Response:** We thank the Reviewer for this suggestion. We rewrote this sentence as follows:

"Most of the climate models do not provide separately the data of evaporation from canopy (i.e.

transpiration) and water evaporation from soil (i.e. evaporation) which complicates attributing changes in evaporation to canopy or soil related processes. Hence, the term 'evaporation' used in this study is referred to both transpiration and evaporation."

• Line 65: "…, CMIP5 data is useful…" Why is the data useful? Because of the abundance of data based on climate models with slightly different assumptions?

**Response:** The CMIP data is useful as they help to better understand past and future climate change, not only because the abundance of data. In fact, all models are not correct and climate models still have great uncertainties. Hence, using the results from multi models are important to reduce these uncertainties.

We rewrote this sentence as follows to clarify this point:

"However, CMIP5 data is useful for better understanding of past and future climate and provides additional understanding about the connections between major climate modes and global evaporation."

We also mentioned the usefulness of climate model simulations in the Introduction as follows:

"As long term and reliable evaporation data is lacking (e.g., Hegerl et al., 2015; Miralles et al.,

2016), climate model simulations provide additional opportunity to examine the impacts of main climate modes on global evaporation."

• Line 73-74: Why focus on the tropical Pacific, tropical Indian and North Atlantic Oceans with the mentioned climate modes? Please motivate why these (and related climate modes) are in focus.

**Response:** We added the following sentence to the Methods section to clarify this point:

"These climate modes are the main sources of global climate variability at interannual timecales (e.g., Abram et al., 2003; Hurrell et al., 2003; McPhaden et al., 2006)."

• Line 79: Please elaborate on what the order of the causal model means.

**Response:** We rewrote this sentence as follows to clarify the meaning of the order:

"…$p \geq 1$ is the order (or the number of lagged time series) of the causal model…"

• Line 90: What is it that makes the applied techniques necessarily robust?

**Response:** We think the applied techniques (i.e., Granger causality test) are well established for detecting causal effects. We rewrote the Methods section as follows to clarify this point:

"We apply test of Granger causality for the predictive model described in equation (1).

Specifically, in order to assess the causal influence from $Y$ to $X$, we compute the probability of the null hypothesis for an absence of Granger causality from $Y$ to $X$. The model shown in equation (1) is defined as a complete predictive model where all variables (i.e., past data of evaporation and climate indices) are used to estimate evaporation. The null model of no causal effects from given climate mode (i.e., variable $Y$) to evaporation is defined by removing the terms related to $Y$ (i.e., by setting $\beta_i = 0 \ with \ i = 1, …, p$) in equation (1). The complete model and the null model are then compared by using the following indicator:

$$L_{Y \to X} = n(\log\left|\Omega_{p,\beta_i=0}\right| - \log\left|\Omega_p\right|)$$

where $|\Omega_p|$ is the determinant of the covariance matrix of the noise residual, and n is the length of the data time series. We test the significance of the complete model by comparing the $L_{Y \to X}$

indicator against a $\chi_p^2$ null distribution. This test results in a probability for no causal effect of the considered variable *Y* on evaporation."

• Line 100-102: "Specifically, high agreement of climate models in teleconnection between

ENSO and tropical ocean evaporation implies that models can simulate the impacts of ENSO on evaporation." Why does a high agreement between models necessarily signal a capability to simulate impacts of ENSO on evaporation?

**Response:** We thank the Reviewer for raising this point. The map of ENSO-evaporation connection presented here (Figures 1 and S1) confirm the results obtained previously (e.g.,

ENSO influence on evaporation of Australia and Amazonia where there is high consistency between models as shown in Figure 1 and S1). Thus, we think this result show the capability of the models. However, there is uncertainty in some other regions.

We removed this sentence to avoid confusing the readers.

• Line 105: Why focus on two significance levels and not just on one? If two significance levels are necessary, please indicate why.

**Response:** The purpose of using two significance levels is only for visualisation. In Figures 2

and 3, the region with 5% significance level is too small (e.g., for the effects of NAO and the

IOD) and we think it would be useful to provide additional information with 10% significance level. In some cases, different significance levels should be used to show different level of uncertainty (e.g., Stocker et al., 2013).

• Line 145: Figure S3 and S4 do not show seasonal responses of evaporation to the IOD. Do you mean Figure S5 and S6?

**Response:** We thank the reviewer for pointing this out. We corrected to Figure S5 and S6 for the seasonal responses of evaporation to the IOD.

• Line 170: The differences in Fig. 4 do not imply unique combinations: a difference of -.1 can result from Granger causalities 0 and .1 or .9 and 1. I would advise to test whether the difference is significant or not.

**Response:** We agree with the reviewer that these differences do not imply unique combinations.
However, we note these values are original and are true values. Figure 4 is only the additional
result of Figures 1, 2 and 3 which show the multi-model mean map of probability for no Granger
causal impact from individual climate mode to global evaporation. The results described in
Figures 1, 2 and 3 are tested for significance.

We note that additional information is added to the Methods section as follows to clarify the
significance test:

"We apply test of Granger causality for the predictive model described in equation (1).
Specifically, in order to assess the causal influence from $Y$ to $X$, we compute the probability of
the null hypothesis for an absence of Granger causality from $Y$ to $X$. The model shown in
equation (1) is defined as a complete predictive model where all variables (i.e., past data of
evaporation and climate indices) are used to estimate evaporation. The null model of no causal
effects from given climate mode (i.e., variable $Y$) to evaporation is defined by removing the
terms related to $Y$ (i.e., by setting $\beta_i = 0 \; with \; i = 1, ..., p$) in equation (1). The complete model
and the null model are then compared by using the following indicator:

$$L_{Y \to X} = n(\log|\Omega_{p,\beta_i=0}| - \log|\Omega_p|)$$

where $|\Omega_p|$ is the determinant of the covariance matrix of the noise residual, and n is the length of
the data time series. We test the significance of the complete model by comparing the $L_{Y \to X}$
indicator against a $\chi_p^2$ null distribution. This test results in a probability for no causal effect of the
considered variable $Y$ on evaporation."

• Line 185-194: Are these results or do these belong in the discussion section?

**Response:** Yes, we think these sentences discuss the connection between the present study and
previous ones (Martens et al., 2018; Miralles et al., 2013).

• Line 209: In Figure 5, panels a and b can be combined, just as panels c and d to be able to be able to visually compare a bit better. The authors could even think of combining all the panels together, to see be able to compare easily between land and ocean, past and future (there are only

12 bars).

**Response:** We thank the reviewer for this advice. We replaced Figure 5 by a new Figure as
follows:

[Figure]

[Figure]

Figure 2. Fraction of Earth surface for land and ocean with probability for the absence of Granger causality between climate
modes and evaporation less than 0.1 (i.e., $p$ value < 0.1). The results are shown for the influence of individual climate mode on
annual mean evaporation for periods 1906-2000 (a) and 2006-2100 (b). Fraction areas influenced by ENSO, NAO and IOD are
shown in blue, red and yellow bars, respectively. Several fraction areas are close to zero. ENSO = El Niño–Southern Oscillation.
NAO = North Atlantic Oscillation. IOD = Indian Ocean Dipole.

• Line 217-220: What is the added value of Figure S10 (showing Figure 5 with p 0.25)? The conclusion remains the same and inferences of this figure aren't mentioned.

**Response:** At standard significance level of 5%, the fraction areas for NAO and IOD are low and close to zero. We think it is important to provide additional information to show that NAO

and the IOD still have the causal effects on evaporation at higher significance level. We also think this Figure S10 makes it easier to compare the effects of NAO and the IOD.

We added the following text to the Discussions Section:

"Figure S10 indicates that the land and ocean area influenced by the IOD is slightly higher compared to NAO."

• Line 231: Same comment as for Figure 4: I would advise to test whether the means are significantly different or not.

**Response:** We thank the Reviewer for pointing this out. Figure 6 is only additional result of

Figures 1, 2 and 3 which show the multi-model mean map of probability for no Granger causal impact from individual climate mode to global evaporation. The results described in Figures 1, 2

and 3 are tested for significance.

• Line 231-232: "Specifically, the fraction area of Earth surface showing lower probability of

ENSO effects for 2006-2100 period is approximately 52.9% (Figure 6a)." I highly doubt this conclusion. The authors should first assess whether the differences are significant or not and after that determine reassess the overall ENSO effects. 52.9% is just slightly higher than just assigning an de(in)crease of ENSO effects randomly (50%). This also goes for following similar conclusions.

**Response:** The differences shown here are original and true values because all the p-values are the results of Granger causality test for all climate models. The value 52.9% is not a random number but it is an original value and it shows the fact that ENSO causal effects on global evaporation are slightly different between the two periods 1906-2000 and 2006-2100 (See also

Figure 1). Figure 6 is only an additional result of Figures 1, 2 and 3 which show the multi-model mean map of probability for no Granger causal impact from individual climate mode to global evaporation. The results described in Figures 1, 2 and 3 are tested for significance.

**Technical corrections**

• Line 40: "… between individual climate mode**s**"

**Response:** We corrected as your suggestion.

• Line 56: "Using other data periods with similar length**s** (i.e., 95 years) **do** not alter…"

**Response:** We corrected this sentence as follows:

"Using other data periods with similar lengths (i.e., 95 years) do not alter the results and conclusions."

• Line 61: "Most of **the** climate models…"

**Response:** We corrected as your suggestion.

• Line 158: "…of NAO impact**s**…"

**Response:** We corrected as your suggestion.

• Line 187: "…in agreement with previous studie**s**…"

**Response:** Here we only cite one study.

• Line 198: "in **the** Australian continent…"

**Response:** We corrected as your suggestion.

• Line 229-230: "…for **the** future period 2006-2100 and **the** historical period 1906-2000…"

**Response:** We corrected as your suggestion.

• Line 264: "…for a short term…"

**Response:** We corrected as your suggestion.

**2 References**

Abram, N. J., Gagan, M. K., McCulloch, M. T., Chappell, J. and Hantoro, W. S.: Coral reef death during the 1997 Indian Ocean Dipole linked to Indonesian wildfires., Science, 301(5635), 952–955, doi:10.1126/science.1094047, 2003.

Hegerl, G. C., Black, E., Allan, R. P., Ingram, W. J., Polson, D., Trenberth, K. E. . . . and Zhang, X.: Challenges in Quantifying Changes in the Global Water Cycle, Bull. Am. Meteorol. Soc., 2015.

Hurrell, J. W., Kushnir, Y., Ottersen, G. and Visbeck, M.: An Overview of the North Atlantic Oscillation, in The North Atlantic Oscillation: Climate Significance and Environmental Impact, pp. 1–35, American Geophysical Union., 2003.

Knutti, R., Furrer, R., Tebaldi, C., Cermak, J. and Meehl, G. a.: Challenges in Combining Projections from Multiple Climate Models, J. Clim., 23(10), 2739–2758, doi:10.1175/2009JCLI3361.1, 2010.

Martens, B., Waegeman, W., Dorigo, W. A., Verhoest, N. E. C. and Miralles, D. G.: Terrestrial evaporation response to modes of climate variability, npj Clim. Atmos. Sci., 1(1), 43, doi:10.1038/s41612-018-0053-5, 2018.

McPhaden, M. J., Zebiak, S. E. and Glantz, M. H.: ENSO as an integrating concept in earth science., Science, 314(December), 1740–1745, doi:10.1126/science.1132588, 2006.

Miralles, D. G., van den Berg, M. J., Gash, J. H., Parinussa, R. M., de Jeu, R. a. M., Beck, H. E., Holmes, T. R. H., Jiménez, C., Verhoest, N. E. C., Dorigo, W. a., Teuling, A. J. and Johannes Dolman, A.: El Niño–La Niña cycle and recent trends in continental evaporation, Nat. Clim. Chang., 4(1), 1–5, doi:10.1038/nclimate2068, 2013.

Miralles, D. G., Jiménez, C., Jung, M., Michel, D., Ershadi, A., Mccabe, M. F., Hirschi, M., Martens, B., Dolman, A. J., Fisher, J. B., Mu, Q., Seneviratne, S. I., Wood, E. F. and Fernández-Prieto, D.: The WACMOS-ET project - Part 2: Evaluation of global terrestrial evaporation data sets, Hydrol. Earth Syst. Sci., 20(2), 823–842, doi:10.5194/hess-20-823-2016, 2016.

Stocker, T. F., D. Qin, G.-K., Plattner, L. V., Alexander, S. K., Allen, N. L., Bindoff, F.-M., Bréon, J. a., Church, U., Cubasch, S., Emori, P., Forster, P., Friedlingstein, N., Gillett, J. M., Gregory, D. L., Hartmann, E., Jansen, B., Kirtman, R., Knutti, K., Krishna Kumar, P., Lemke, J.,

Marotzke, V., Masson-Delmotte, G. a., Meehl, I. I., Mokhov, S., Piao, V., Ramaswamy, D.,

Randall, M., Rhein, M., Rojas, C., Sabine, D., Shindell, L. D., Talley, D. G. and Xie, V. and S.-

P.: Technical Summary, Clim. Chang. 2013 Phys. Sci. Basis. Contrib. Work. Gr. I to Fifth

Assess. Rep. Intergov. Panel Clim. Chang., 33–115, doi:10.1017/ CBO9781107415324.005,

2013.

Weigel, A. P., Knutti, R., Liniger, M. a. and Appenzeller, C.: Risks of Model Weighting in

Multimodel Climate Projections, J. Clim., 23(15), 4175–4191, doi:10.1175/2010JCLI3594.1,

2010.

---

## Referee Report (RR1)

HESS-2019-442

**Le** *et al.* **(2019):** Response of global evaporation to major climate modes in historical and future CMIP5 simulations

**GENERAL COMMENTS**

I have the feeling that most of my comments have been properly tackled and that the article is close to publication-proof. I still have the feeling that the authors miss some chances to truly make this a high-impact paper. For instance, I think that elaborating more on the differences between the "current" and "future" impact of climate modes on evaporation is very interesting. However, I do see that this would need substantially more work: building up in-depth knowledge of the different climate models, running extra analyses, and physically interpreting the results. Below, I list some final comments that need to be tackled before the paper can be published.

**SPECIFIC COMMENTS**

1. It is still not clear to me from the description at P2-L61–64 why the historical period and future period need to cover the same number of years. As the methods used in this study have a statistical nature, I would say that more data gives more robust results (and conclusions). The authors also mention that "periods with similar lengths do not alter the results". Why is the latter statement of importance then?

2. It is not clear why the authors de-trend their data. At P4-L103, the authors claim that this processing step does not change the results. When this step has no impact on the analysis, why is it performed then? I think a motivation should be given in the manuscript to avoid confusing the reader.

3. The description of the temporal resolution at P4-L125–130 is very vague and unclear to me. Please revise this description to make it more readable.

4. I am still surprised about the low impact of the climate modes on evaporation dynamics over land areas. Previous numbers reported in literature are generally higher than the ones reported in this study (P8-L253–266). I can agree that different methods give different results, but the authors should be able to explain these differences when they are that substantial. This difference should at least be acknowledged and better discussed in the manuscript by contrasting the numbers against similar values reported in literature.

5. The statement at P9-L259 is interesting: "We observe an increase in land area affected by ENSO to …". Why do the authors not elaborate a bit more on this? Why do they think they see this increase? The nice feature of this study is that a separate analysis is performed for a "current" and "future" period. In my opinion, elaborating a bit more on the different results between both periods would really make this a stronger paper.

6. The discussion at P9-L267–274 is interesting as well. I think another reason might be that the modes are affecting meteorological variables that (during that period) are not driving evaporation. E.g. NAO might affect precipitation (water availability) in northern Europe during winter, but in wintertime, evaporation is mainly driven by radiation in this region, so NAO will apparently only have a small (or no) impact on the dynamics of evaporation in that case.

**TECHNICAL CORRECTIONS**

1. P2-L49: this is the first use of "CMIP" in the main text, the abbreviation should thus be defined here (and not at P2-L54).

2. P3-L73–74 : this is a repetition of P2-L31.

3.  P3-L68–69: this description can be easily miss-interpreted as this is not how the authors deal with the multi-model ensemble (details are described at P4-L123–124) in this study. I would suggest rephrasing this statement.

4.  P4-L134: "… influence on evaporation of different regions …": please rephrase because it is not clear what the authors mean.

5.  P4-L140: "… which is the main contributor of change in …": the authors use "change" multiple times along these lines, but to me, it is not exactly clear what the authors mean with "change". I guess they mean that dynamics of evaporation are affected/changed by something else, but it sounds vague to me.

*Brecht Martens*
*Ghent University*
*Laboratory of Hydrology and Water Management*

---

## Author Response (AR2)

**Response to Reviewer #1's comments**

HESS-2019-442

Le et al. (2019): Response of global evaporation to major climate modes in historical and future CMIP5 simulations

5   GENERAL COMMENTS

I have the feeling that most of my comments have been properly tackled and that the article is close to publication-proof. I still have the feeling that the authors miss some chances to truly make this a high-impact paper. For instance, I think that elaborating more on the differences between the "current" and "future" impact of climate modes on evaporation is very interesting. However, I do see that this would
10  need substantially more work: building up in-depth knowledge of the different climate models, running extra analyses, and physically interpreting the results. Below, I list some final comments that need to be tackled before the paper can be published.

**Response:** We thank the Reviewer for these suggestions. We modified the manuscript based on your suggestions as below.

15   SPECIFIC COMMENTS

1. It is still not clear to me from the description at P2-L61–64 why the historical period and future period need to cover the same number of years. As the methods used in this study have a statistical nature, I would say that more data gives more robust results (and conclusions). The authors also mention that "periods with similar lengths do not alter the results". Why is the latter statement of
20  importance then?

**Response:** Because we directly compare the results (i.e. the probability map for the absence of Granger causality between climate modes and evaporation) of the historical period and future period, it is safe to use the same years for both periods. Depending on each dataset, data length may affect the statistical significance (i.e., probability) of the results. Thus, we think it is better to use similar length for both
25  periods (particularly when the probabilities are compared).

We removed the sentence "Using other data periods with similar lengths (i.e., 95 years) do not alter the results and conclusions" to avoid confusing the reader.

2. It is not clear why the authors de-trend their data. At P4-L103, the authors claim that this processing step does not change the results. When this step has no impact on the analysis, why is it performed then? I think a motivation should be given in the manuscript to avoid confusing the reader.

**Response:** We detrend the data for the purpose of simplifying and standardizing the data. We rewrote this sentence as follows to clarify this point:

"Detrending the data (for standardizing) does not alter the results and conclusions."

3. The description of the temporal resolution at P4-L125–130 is very vague and unclear to me. Please revise this description to make it more readable.

**Response:** We rewrote these sentences as follow:

"We note that the temporal resolution of all analyses is yearly (i.e., the index $t$ in equation (1) denotes year). Although the definition of the predictand (i.e., $X_t$ in equation (1)) is either annual mean or seasonal mean values, the analyses related to Granger causality are computed on annual basis. We report the analyses using the annual mean of evaporation (i.e., the predictand $X_t$) as the main results of this study (Figures 1-7) while the analyses using the seasonal mean of evaporation provide additional information (Figures S3-4, S6-7, S9-10)."

4. I am still surprised about the low impact of the climate modes on evaporation dynamics over land areas. Previous numbers reported in literature are generally higher than the ones reported in this study (P8-L253–266). I can agree that different methods give different results, but the authors should be able to explain these differences when they are that substantial. This difference should at least be acknowledged and better discussed in the manuscript by contrasting the numbers against similar values reported in literature.

**Response:** We agree with the Reviewer. We added the following sentences to clarify this point:

"The influence of climate modes on land evaporation shown in the present study is generally lower than the results reported in previous works (e.g., Martens et al., 2018). This difference is due to the use of different methods and different data periods."

5. The statement at P9-L259 is interesting: "We observe an increase in land area affected by ENSO to …". Why do the authors not elaborate a bit more on this? Why do they think they see this increase? The nice feature of this study is that a separate analysis is performed for a "current" and "future" period. In my opinion, elaborating a bit more on the different results between both periods would really make this a stronger paper.

**Response:** We think the reason for this increase might be the increase in frequency of extreme ENSO events in the future period (Cai et al., 2015). However, we also think this topic could be an open question that requires different approach to answer.

We tried to elaborate a bit more on this by adding the following sentence:

"The increase of ENSO effects on land evaporation might be associated with the increase in frequency of extreme ENSO events in the future period (Cai et al., 2015)"

6. The discussion at P9-L267–274 is interesting as well. I think another reason might be that the modes are affecting meteorological variables that (during that period) are not driving evaporation. E.g. NAO might affect precipitation (water availability) in northern Europe during winter, but in wintertime, evaporation is mainly driven by radiation in this region, so NAO will apparently only have a small (or no) impact on the dynamics of evaporation in that case.

**Response:** We thank the Reviewer for this additional clarification. We added the following sentences to this discussion:

"Besides, climate modes may affect meteorological variables that do not drive land evaporation dynamics. For instance, NAO might affect precipitation (water availability) in northern Europe during winter (Hurrell et al., 2003), but in wintertime, land evaporation is mainly driven by solar radiation in this region. Thus, NAO will apparently only have a small (or no) impact on the dynamics of land evaporation in that case."

TECHNICAL CORRECTIONS

1. P2-L49: this is the first use of "CMIP" in the main text, the abbreviation should thus be defined here (and not at P2-L54).

**Response:** We thank the Reviewer for pointing this out. We rewrote these sentences as your suggestion.

80  2. P3-L73–74: this is a repetition of P2-L31.

Response: We removed the sentence "Hence, the term 'evaporation' used in this study is referred to both transpiration and evaporation" to avoid the repetition.

3. P3-L68–69: this description can be easily miss-interpreted as this is not how the authors deal with the multi-model ensemble (details are described at P4-L123–124) in this study. I would suggest rephrasing

85  this statement.

Response: We rewrote this sentence as follows:

"The results based on multi-model mean are generally better and more reliable than single model results (e.g., Weigel et al., 2010)."

4. P4-L134: "… influence on evaporation of different regions …": please rephrase because it is not

90  clear what the authors mean.

Response: We rewrote this sentence as follows:

"…ENSO is more likely to have influence on numerous regions (highlighted in brown shades) of both hemispheres…"

5. P4-L140: "… which is the main contributor of change in …": the authors use "change" multiple

95  times along these lines, but to me, it is not exactly clear what the authors mean with "change". I guess they mean that dynamics of evaporation are affected/changed by something else, but it sounds vague to me.

Response: We rewrote this sentence as follows:

"…which is the main contributor to variations of evaporation…"

100  *Brecht Martens*

*Ghent University*

*Laboratory of Hydrology and Water Management*

**Response to Reviewer #2's comments**

Submitted on 08 Jan 2020

105 Referee #2: Jasper Denissen, jdenis@bgc-jena.mpg.de

1) While I do like the addition of Figure 7, I'm not satisfied line 290 - 293. This figure allows to elaborate more on where and why there are the largest differences between dominant climate modes or where and why there are no differences. If the differences between past and future are not mentioned, why show both past and future at all? A shift from one dominant climate mode to another means a shift
110 in dynamics related to their respective climate modes, which could change the timing and magnitude of evaporation.

**Response:** We thank the Reviewer for the comment. We added the following sentences to further discuss the results shown in Figure 7:

"This result indicates the important role of ENSO on global evaporation with dominant effects in
115 tropical Pacific and parts of middle Asia, Indochina Peninsula, Australia and northeastern South America. The IOD has dominant effects in western tropical Indian Ocean and small part of eastern tropical Pacific while NAO is a dominant mode in the North Atlantic and European regions. The shift from one dominant climate mode to another indicates a shift in the dynamics related to their respective climate modes, which could change the timing and magnitude of evaporation. There is minor change of
120 the dominance of climate modes between historical and future periods, with slight reduction in the dominance of the IOD in the future."

2) I appreciate the addition of line 104 - 113, but I still don't have an idea about what makes this technique robust (line 114 - 115). Maybe I should ask what you mean with robust? In my opinion, robustness means for example bootstrapping the analysis and getting similar results.

125 **Response:** We mean the result of causal influence obtained by using this technique is not spurious. Here, the assessment is robust mainly because the method is designed to detect causal effects (i.e., as shown in the predictive model in equation 1 and the significance test described in the Methods section). This technique is based on a well-established method which is used in numerous studies (e.g., Le et al., 2016; Mosedale et al., 2006; Pasini et al., 2012; Stern and Kaufmann, 2013). To our knowledge, in
130 order to obtain robust conclusions of causal connection, bootstrapping in normally necessary for other

methods (e.g., correlation analysis) because these methods are not particularly designed to detect causal relationship. Besides, the result is also supported by multi-models approach.

We rewrote this sentence as follows to clarify this point:

"The techniques employed in the present study, which are designed to detect causal relationship, provide robust assessment about the causal influence of considered climate mode on global evaporation."

3) The significance of the results in Figures 1, 2 and 3 is different from the significance of the results shown in Figure 4. One could think of it this way: if for example the probability for the absence of Granger causality between ENSO and annual mean evaporation for the period 1906-2000 is 0.1 and significant (as noted by the green and red contour lines in Figure 1) and the probability for the absence of Granger causality between IOD and annual mean evaporation for the period 1906-2000 is 0.1 and significant, the difference between these two is approximately 0, which makes the difference insignificant. What I'm getting at is that you need another significance test to test whether the difference is significantly different from 0, which could be achieved with a two-sample t-test. This should be done for Figure 4 and 6 and might alter the percentages mentioned in the Discussions section.

**Response:** We thank the Reviewer for raising this interesting point. We understand the approach pointed out by the Reviewer. However, we think the difference of probability shown in Figure 4 should not be a target for a significance test. In our opinion, these differences are original and true values, meaning that we receive only one value for each grid point and accept as it is. We also emphasize that the reported differences are the multi-model results and the uncertainties are reduced.

**References**

Cai, W., Wang, G., Santoso, A., McPhaden, M. J., Wu, L., Jin, F.-F., Timmermann, A., Collins, M., Vecchi, G., Lengaigne, M., England, M. H., Dommenget, D., Takahashi, K. and Guilyardi, E.: Increased frequency of extreme La Niña events under greenhouse warming, Nat. Clim. Chang., 5(2), 132–137, doi:10.1038/nclimate2492, 2015.

Le, T., Sjolte, J. and Muscheler, R.: The influence of external forcing on subdecadal variability of

regional surface temperature in CMIP5 simulations of the last millennium, J. Geophys. Res. Atmos., 121(4), 1671–1682, doi:10.1002/2015JD024423, 2016.

Martens, B., Waegeman, W., Dorigo, W. A., Verhoest, N. E. C. and Miralles, D. G.: Terrestrial
160  evaporation response to modes of climate variability, npj Clim. Atmos. Sci., 1(1), 43, doi:10.1038/s41612-018-0053-5, 2018.

Mosedale, T. J., Stephenson, D. B., Collins, M. and Mills, T. C.: Granger causality of coupled climate processes: Ocean feedback on the North Atlantic Oscillation, J. Clim., 1182–1194 [online] Available from: http://journals.ametsoc.org/doi/abs/10.1175/JCLI3653.1 (Accessed 17 January 2014), 2006.

165  Pasini, A., Triacca, U. and Attanasio, A.: Evidence of recent causal decoupling between solar radiation and global temperature, Environ. Res. Lett., 7(3), 034020, doi:10.1088/1748-9326/7/3/034020, 2012.

Stern, D. I. and Kaufmann, R. K.: Anthropogenic and natural causes of climate change, Clim. Change, 122(1–2), 257–269, doi:10.1007/s10584-013-1007-x, 2013.

**Response of global evaporation to major climate modes in historical and future CMIP5 simulations**

Thanh Le[1, 2], Deg-Hyo Bae[1, 2]

[1]Department of Civil & Environmental Engineering, Sejong University, Seoul, South Korea
[2]Center for Climate Change Adaptation for Water Resources, Sejong University, Seoul, South Korea

*Correspondence to*: Deg-Hyo Bae (dhbae@sejong.ac.kr)

**Abstract.** Climate extremes, such as floods and droughts might have severe economic and societal impacts. Given the high costs associated with these events, developing early warning systems are of high priority. Evaporation, which is driven by around 50% of solar energy absorbed at surface of the Earth, is an important indicator of global water budget, monsoon precipitation, drought monitoring and hydrological cycle. Here we investigate the response of global evaporation to main modes of interannual climate variability, including the Indian Ocean Dipole (IOD), the North Atlantic Oscillation (NAO) and the El Niño-Southern Oscillation (ENSO). These climate modes may have influence on temperature, precipitation, soil moisture, wind speed, and are likely to have impacts on global evaporation. We utilized data of historical simulations and RCP8.5 future simulations derived from Coupled Model Intercomparison Project Phase 5 (CMIP5). Our results indicate that ENSO is an important driver of evaporation for many regions, especially the tropical Pacific. The significant IOD influence on evaporation is limited in western tropical Indian Ocean while NAO is more likely to have impacts on evaporation of the North Atlantic European areas. There is high agreement between models in simulating the effects of climate modes on evaporation of these regions. Land evaporation is found to be less sensitive to considered climate modes compared to oceanic evaporation. The spatial influence of major climate modes on global evaporation is slightly more significant for NAO and the IOD while slightly less significant for ENSO in the 1906-2000 period compared to the 2006-2100 period. This study allows us to obtain insight about the predictability of evaporation, and hence, may improve the early warning systems of climate extremes and water resources management.

**Keywords:** ENSO; IOD; NAO; Evaporation; CMIP5; Water resources management;

**1    Introduction**

The North Atlantic Oscillation (NAO; e.g., Hurrell et al., 2003), the Indian Ocean Dipole (IOD; Saji et al., 1999; Webster et al., 1999), and the El Niño-Southern Oscillation (ENSO; e.g., Bjerknes, 1969; Neelin et al., 1998) are major modes of global climate variability. These climate modes may have influence on important drivers of evaporation such as surface temperature (e.g., Arora et al., 2016; Leung & Zhou, 2016; Sun et al., 2016; Thirumalai et al., 2017; Wang et al., 2017), precipitation (Dai and Wigley, 2000), soil moisture (Nicolai-Shaw et al., 2016), humidity (Hegerl et al., 2015) and wind speed (Hurrell et

al., 2003; Yeh et al., 2018). Hence, these climate modes are likely to have impacts on global evaporation and transpiration (hereafter simply referred as 'evaporation'). Evaporation, which is driven by around 50% of solar energy absorbed at surface of the Earth (Cavusoglu et al., 2017; Jung et al., 2010), is an important variable contributing to global water budget, monsoon precipitation, drought monitoring and hydrological cycle (Friedrich et al., 2018; Kitoh, 2016; Lee et al., 2019; Van Osnabrugge et al., 2019; Son and Bae, 2015). Additionally, changes in global evaporation are expected to feedback on global and regional climate. For example, land evaporation is shown to have influences on carbon cycles (Cheng et al., 2017), cloud cover (Teuling et al., 2017) and air temperature (Miralles et al., 2012).

While previous studies emphasize the importance of ENSO (Martens et al., 2018; Miralles et al., 2013), the Atlantic Multidecadal Oscillation, the Tropical Northern Atlantic Dipole, Tropical Southern Atlantic Dipole and the IOD (Martens et al., 2018) on global land evaporation, the role of the NAO remains elusive. In addition, the future influence of these climate modes on global evaporation under warming environment remain unclear. Climate change and rising temperature might increase surface evaporation (Miralles et al., 2013), and thus, might reduce global water availability and cause change in hydrological cycle (e.g., Naumann et al., 2018). Moreover, most of previous works (e.g., Shinoda and Han, 2005; Xing et al., 2016; Zveryaev and Hannachi, 2011) mainly address the connection between individual climate modes and evaporation, however, the role of other climate modes might not be included in the analyses. As long term and reliable evaporation data is lacking (e.g., Hegerl et al., 2015; Miralles et al., 2016), climate model simulations provide additional opportunity to examine the impacts of main climate modes on global evaporation. Besides, evaluating the models' consistency in reproducing the impacts of internal climate variability on evaporation is important for understanding the difference between models.

Here we investigate the causal impacts of major climate modes (i.e. ENSO, IOD and NAO) on global terrestrial and oceanic evaporation in Coupled Model Intercomparison Project Phase 5 (CMIP5)CMIP5 model simulations for the 1906-2000 and the 2006-2100 periods. For this investigation, we use multivariate predictive models and tests of Granger causality which consider the simultaneous impact of climate modes on global evaporation (see Methods section 2.2).

**2    Data and Methods**

**2.1    Data**

The data used in the present study was obtained from CMIP5 model simulations. We employ data of historical simulations (experiment name 'historical' in CMIP5) and future simulations with high-emissions climate change scenario (experiment name Representative Concentration Pathway [RCP] 8.5 scenario, 'rcp85') (Taylor et al., 2012). The RCP8.5 is a very high emission scenario with radiative forcing of 8.5 W/m$^2$ in 2100 relative to the preindustrial level (van Vuuren et al., 2011). Warming environment in the RCP8.5 scenario increases the frequency of extreme ENSO and IOD events (Cai et al., 2014, 2015) and potentially modulates the impacts of these climate modes on global evaporation. The starting year of historical simulations is roughly 1850 and the ending year is roughly 2005 while the starting year of future simulations is roughly 2006 and the ending year is roughly 2100. The results of the effects of

climate modes on evaporation are compared between the historical period and the future period. Hence, in our analyses, we only use the data for the 1906-2000 historical period as a reference (with similar data length) for the future period 2006-2100.  We use different data variables, including monthly sea level pressure (i.e., 'psl' in CMIP5 datasets), sea surface temperature (i.e., 'ts'), and evaporation (i.e., 'evspsbl'). For each model, we only utilize one simulation (i.e., 'r1i1p1'). The models employed in this study are listed in Table S1 (Supplement). As we use 15 different models for our analysis, the uncertainties related to the effects of climate modes on evaporation are reduced. The results based on multi-model mean  are generally better and more reliable than single model results (e.g., Weigel et al., 2010). Terrestrial evaporation is the flux of water at surface into the atmosphere due to transformation of both solid and liquid phases to vapor (from vegetation and underlying surface). Most of the climate models do not provide separately the data of evaporation from canopy (i.e. transpiration) and water evaporation from soil (i.e. evaporation) which complicates attributing changes in evaporation to canopy or soil related processes.

There might exist model biases in simulating ENSO (e.g. Taschetto et al., 2014), the IOD (Chu et al., 2014; Weller and Cai, 2013), NAO (Gong et al., 2017; Lee et al., 2018) and there is uncertainty in capability of land surface models in modeling evaporation (Mueller & Seneviratne, 2014; Wang & Dickinson, 2012). However, CMIP5 data is useful for better understanding of past and future climate and provides additional understanding about the connections between major climate modes and global evaporation.

**2.2    Methods**

The NAO index (Hurrell et al., 2003) is computed as the first empirical orthogonal function (EOF) of boreal winter (DJF) sea level pressure (SLP) anomalies in the North Atlantic area (90ºW-40ºE, 20º-70ºN). We compute the dipole mode index (DMI) (Saji et al., 1999) as the discrepancy of SST anomalies between the western tropical Indian Ocean (50ºE-70ºE; 10ºN-10ºS) and the south-eastern tropical Indian Ocean (90ºE-110ºE; 0ºN-10ºS) in the boreal fall (SON). We define the ENSO index as the average sea surface temperature (SST) anomalies in the Niño 3.4 region (120°W-170°W; 5Nº-5ºS) during boreal winter. We include ENSO, IOD and NAO in our analyses as these indices are three major climate modes of tropical Pacific, tropical Indian and North Atlantic Oceans. These climate modes are the main sources of global climate variability at interannual timecales (e.g., Abram et al., 2003; Hurrell et al., 2003; McPhaden et al., 2006).

We evaluate the causal effects of a climate mode (i.e., NAO or DMI or ENSO) on evaporation by using the following predictive model (e.g., Mosedale et al., 2006):

$$X_t = \sum_{i=1}^{p} \alpha_i X_{t-i} + \sum_{i=1}^{p} \beta_i Y_{t-i} + \sum_{j=1}^{m} \sum_{i=1}^{p} \delta_{j,i} Z_{j,t-i} + \varepsilon_t \tag{1}$$

where $X_t$ is the annual mean (or seasonal mean) evaporation for year $t$, $Y_t$ is the selected index (i.e., ENSO or NAO or DMI) for evaluating the causal effects on evaporation for year $t$, $Z_{j,t}$ is the confounding variable $j$ for year $t$, $p \geq 1$ is the order (or

the number of lagged time series) of the causal model, and $m$ is total number of confounding variables. The optimal order $p$ is computed by minimizing the Bayesian information criterion or Schwarz criterion (Schwarz, 1978). We note that the optimal orders might be different for each grid cell, depending on evaporation data of the selected grid cell. The optimal

265 orders are normally less than 8 in our analysis, suggesting that the impact of major climate modes on evaporation is evaluated at inter-annual time scales. Confounding variables (e.g., if NAO is the selected index of possible causal influence on evaporation, the confounding variables are DMI and ENSO) may have impacts on the links between selected index and global evaporation. There are two form of confounding variables in our analysis, hence, $m$ is equal to 2 in equation (1). The regression coefficients $\alpha_i$, $\beta_i$ and $\delta_{j,i}$ and the noise residuals $\varepsilon_t$ are computed by using multiple linear regression analysis and

270 least squares method. All climate indices and evaporation data are normalized (by using z-score) and detrended (by subtracting the trending line from given data; the trending line or the best-fit line is identified using least squares method). Detrending the data (for standardizing) does not alter the results and conclusions.

We apply test of Granger causality for the predictive model described in equation (1). Specifically, in order to assess the causal influence from $Y$ to $X$, we compute the probability of the null hypothesis for an absence of Granger causality from $Y$

275 to $X$. The model shown in equation (1) is defined as a complete predictive model where all variables (i.e., past data of evaporation and climate indices) are used to estimate evaporation. The null model of no causal effects from given climate mode (i.e., variable $Y$) to evaporation is defined by removing the terms related to $Y$ (i.e., by setting $\beta_i = 0$ $with$ $i = 1, \dots, p$) in equation (1). The complete model and the null model are then compared by using the following indicator:

$$L_{Y \to X} = n(\log \left| \Omega_{p, \beta_i = 0} \right| - \log \left| \Omega_p \right|)$$

280 where $|\Omega_p|$ is the determinant of the covariance matrix of the noise residual, and n is the length of the data time series. We test the significance of the complete model by comparing the $L_{Y \to X}$ indicator against a $\chi_p^2$ null distribution. This test results in a probability for no causal effect of the considered variable $Y$ on evaporation. Additional information on the test of Granger causality are explained in earlier works (e.g., Le, 2015; Le et al., 2016). The techniques employed in the present study, which are designed to detect causal relationship, provide robust assessment about the causal influence of considered climate mode

285 on global evaporation. In addition, these approaches account for the concurrent influence of confounding variables and hence, provide more realistic evidence of the response of global evaporation to major climate modes.

Modes of climate variability might be correlated to each other and this correlation might have effects on the relationship between these modes and other variables (e.g., evaporation) (Gonsamo et al., 2016; Martens et al., 2018). However, in the approach of Granger causal analysis, the conclusion for the causal effects from variable $Y$ (i.e., the considered climate mode)

290 to variable $X$ (i.e., evaporation) is independent from the relationship between $Y$ and other factors (i.e., the relationship between climate modes) (Mosedale et al., 2006; Stern and Kaufmann, 2013).

We apply the methods described above to all the single models. We then rescale the results of single model to 1° longitude ×1° latitude spatial resolution. We use the rescaled results to compute the multi-model mean which is shown as a map of probability for no Granger causal impact from individual climate mode to global evaporation.

295  We note that the temporal resolution of all analyses is yearly (i.e., the index $t$ in equation (1) denotes year). Although the definition of the predictand (i.e., $X_t$ in equation (1)) is either  annual mean  or seasonal mean values, the analyses related to Granger causality are computed on annual basis. We report the analyses using the annual mean of evaporation (i.e., the predictand $X_t$) as the main results of this study (Figures 1-7) while the analyses using

300  the seasonal mean of evaporation provide additional information (Figures S3-4, S6-7, S9-10).

**3    Results and discussions**

**ENSO influence on evaporation**

The probability maps of no Granger causality between ENSO and global evaporation for the periods 1906-2000 and 2006-2100 are shown in Figure 1. In both periods, ENSO is more likely to have influence on numerous

305  regions (highlighted in brown shades) in both hemispheres, including middle Asia (regions closed to Caspian Sea, details are shown in Figure S1a in the Supplement), Indian Ocean, Indochina Peninsula, Australia (Figure S1b), tropical Pacific and northeastern South America (i.e. Amazonia, Figure S1c) and the Pacific coast of America (Figure S1d). There is high agreement between models (indicated by stippling in Figure 1) in simulating ENSO-evaporation connection of these regions. ENSO might indirectly influence global evaporation by modulating regional climate factors associated with evaporation

310  processes. For example, ENSO significantly influence near surface wind, which is the main contributor of  variations of evaporation (Xing et al., 2016). The ENSO impacts on large part of tropical Pacific Ocean are robust at 5% and 10% significance levels (here we reject the null hypothesis of the absence of Granger causal effects between ENSO and evaporation at 5% and 10% significance levels, hence, we conclude that there is significant causal impact; we note that the 5% and 10% significance levels are computed from the test for the absence of Granger causality). Further analyses reveal

315  that ENSO has significant impacts on SST (Figure S2a) and zonal winds (Figure S2b) over the tropical Pacific for the 1906-2000 period (similar patterns are observed for the 2006-2100 period, not shown). Hence, the influence of ENSO on evaporation might be associated with Wind-Evaporation SST (WES) effect (Cai et al., 2019). The WES effect occurs when warm (cold) water becomes warmer (colder) due to decrease (increase) in evaporation and weakened (strengthened) surface winds. Besides, ENSO is known to induce changes in global precipitation with decrease in rainfall in Africa, Indochina

320  Peninsula, Indonesia, Australia and Amazonia during El Niño phase (Dai and Wigley, 2000) and thus indirectly influence evaporation of these regions. ENSO causal effects on global precipitation are shown in Figure S2c which indicates the close connection between precipitation and evaporation process in several regions (e.g., tropical Pacific, Australia, Amazonia and regions closed to Caspian Sea).

[revised manuscript text omitted]

420  part of Amazonia (Mueller and Seneviratne, 2014). Thus, these biases contribute to the uncertainties in the effects of climate modes on evaporation. Nevertheless, the methods based on multi-model mean and Granger causality tests (see Section 2.2) help to reduce the uncertainties and provide robust results and conclusions.

Figure 5 shows the fraction area of Earth surface for land and ocean with probability for the absence of Granger causality between climate modes and evaporation less than 0.1 (i.e. p value < 0.1; here, the null hypothesis of no Granger causality from climate modes to evaporation is rejected at 10% significance level, hence, we conclude that there is significant causal effects; we note that the fraction area is substantially smaller if p value < 0.05). Specifically, during the period 1906-2000, nearly 1.039% of land area is affected by ENSO at 10% significance level while the affected land areas by NAO and the IOD are 0% and 0.017%, respectively (Figure 5a). The area of oceanic evaporation influenced by ENSO, NAO and the IOD are 2.908%, 0.01% and 0.196%, respectively (Figure 5a). We observe an increase in land area affected by ENSO to 1.38% during the 2006-2100 period while the affected land areas by NAO and the IOD are 0% (Figure 5b). The increase of ENSO effects on land evaporation might be associated with the increase in frequency of extreme ENSO events in the future period (Cai et al., 2015). The area of oceanic evaporation influenced by ENSO, NAO and the IOD are 2.944%, 0.003% and 0.122%, respectively (Figure 5b). This result shows a minor decrease in NAO and IOD effects on oceanic evaporation during the 2006-2100 period compared to the 1906-2000 period. Figure S11 shows additional analyses for the fraction area of Earth surface for land and ocean with probability for the absence of Granger causality between climate modes and evaporation less than 0.25 (i.e., climate modes are unlikely to have no causal effects on evaporation; Stocker et al., 2013). Figure S11 indicates that the land and ocean area influenced by the IOD is slightly higher compared to NAO. The influence of climate modes on land evaporation shown in the present study is generally lower than the results reported in previous works (e.g., Martens et al., 2018). This difference is due to the use of different methods and different data periods.

The considered climate modes (i.e. ENSO, NAO and IOD) are more likely to have influences on global evaporation over oceans while they have limited signature in change of land evaporation for many regions (Figures 1, 2, 3, 5 and S11). These results indicate the role of other factors in modulating land evaporation. Particularly, the influence of major climate modes on land evaporation might be offset by other factors like greenhouse gases, aerosols or solar radiation (Dong and Dai, 2017; Hegerl et al., 2015; Liu et al., 2011). Besides, climate modes may affect meteorological variables that do not drive land evaporation dynamics. For instance, NAO might affect precipitation (water availability) in northern Europe during winter (Hurrell et al., 2003), but in wintertime, land evaporation is mainly driven by solar radiation in this region. Thus, NAO will apparently only have a small (or no) impact on the dynamics of land evaporation in that case. The impacts of climate modes on ocean evaporation contribute to change in global hydrological cycle as ocean evaporation might affect land water cycle by inducing change in regional precipitation (Diawara et al., 2016). For example, the evaporation of the eastern North Pacific is the main moisture supply for precipitation in California (Wei et al., 2016).

Changes in the spatial influences of major modes of climate variability on regional evaporation for the future period 2006-2100 and the historical period 1906-2000 depend on each climate mode (Figures 1, 2, 3). Analyses in details of the difference between these two periods are shown in Figure 6. Specifically, the fraction area of Earth surface showing lower probability of ENSO effects for 2006-2100 period is approximately 52.9% (Figure 6a). This result indicates that ENSO slightly expand the impacted regions (highlighted in red shades, Figure 6a) during 2006-2100 period compared to 1906-2000 period. Conversely, the fraction area of Earth surface for effects of NAO and the IOD during the 2006-2100 period are

decreased with 47.2% and 45.7%, respectively (Figure 6b and 6c). These results suggest that, for several regions of declining impacts of climate modes (highlighted in blue shades, Figure 6), the important drivers of evaporation processes in the 21$^{st}$ century (e.g., precipitation, near-surface air temperature, wind speed, soil moisture) tend to be not affected by the modes of climate variability in the models. For example, response of regional evaporation to climate warming depends on precipitation (Parr et al., 2016; Zhang et al., 2018) and projected rise of surface temperature is shown to mainly contribute to the increase in regional evaporation (Laîné et al., 2014). Because the volume of moisture carried by air increases with air temperature, the atmospheric water vapor demand is expected to increase with rising air temperature and rising greenhouse gases concentration (Miralles et al., 2013). In addition, the declines in pan evaporation in southern/western Australia are mainly caused by decreases in wind speeds (Stephens et al., 2018).

The dominance of an individual climate modes on evaporation is summarized in Figure 7 for historical and future periods. Figure 7 shows the regions where the lowest probability for the absence of Granger causality between climate modes and evaporation is less than 0.25 (i.e., climate modes are unlikely to have no causal effects on evaporation). This result indicates the important role of ENSO on global evaporation with dominant effects in tropical Pacific and parts of middle Asia, Indochina Peninsula, Australia and northeastern South America. The IOD has dominant effects in western tropical Indian Ocean and small part of eastern tropical Pacific while NAO is a dominant mode in the North Atlantic and European regions. The shift from one dominant climate mode to another indicates a shift in the dynamics related to their respective climate modes, which could change the timing and magnitude of evaporation. There is minor change of the dominance of climate modes between historical and future periods, with slight reduction in the dominance of the IOD in the future.

**4    Conclusions**

The CMIP5 historical and RCP8.5 future simulations provide an opportunity to assess the influence of major climate modes on global evaporation, which plays an important role in hydrological cycle, drought monitoring and water resources management. This paper employed tests of Granger causality and showed vigorous evaluation of possible impacts of NAO, the IOD and ENSO on global evaporation.

The results show that ENSO is likely to have impacts on evaporation of different regions in both hemispheres, including tropical Pacific and Indian Oceans, Indochina Peninsula, middle Asia (regions closed to Caspian Sea), Australia, northeastern South America (i.e. Amazonia) and the Pacific coast of North and South America. The impacts of NAO are mainly found in the North Atlantic and European regions while the notable influence of the IOD is limited in western tropical Indian Ocean and part of eastern tropical Pacific. There is high agreement between models in simulating the effects of climate modes on evaporation of these regions. Despite more extreme IOD events are expected in the future (Cai et al., 2013, 2014), the spatial influences of the IOD on evaporation are slightly less significant in the 2006-2100 period compared to the 1906-2000 period. These results imply that the effects on evaporation of the extreme states of the IOD do not persist long enough to be significant. Moreover, in the climate system, the effects of these extreme IOD events might be

compensated by the extreme events of other climate modes (e.g., ENSO). The weak impacts of ENSO, NAO and the IOD on evaporation of several regions suggest the importance of external forcings (e.g. greenhouse gases radiative forcing, solar forcing) and other climate modes on global evaporation variability. We emphasize the strong connection between considered climate modes (i.e. ENSO, IOD and NAO) and oceanic evaporation at interannual time scales. Land evaporation is shown to have weak connection with teleconnection indices in several regions, suggesting the weak effects of climate modes on important drivers of land evaporation, such as local wind speed (Stephens et al., 2018), surface temperature (Laîné et al., 2014; Miralles et al., 2013), moisture supply (Jung et al., 2010) and amount of precipitation (Parr et al., 2016).

Our results may have suggestions for the predictability of regional evaporation (e.g. Australia, tropical Pacific, tropical Indian and North Atlantic Oceans, the Pacific coast of North and South America, Amazonia, Europe, Indochina Peninsula and middle Asia) by using past time series of major climate modes for a short term of several years. The results of this study might provide information for drought and flood prediction as evaporation is an important metric for quantifying drought (McEvoy et al., 2016) and flood events.

Uncertainty regarding the impact of major climate modes on evaporation of several regions (e.g. ENSO impacts on evaporation of South Asia, South Africa, eastern North America, southern South America; the IOD impacts on western Africa and South Asia; NAO impacts on North Atlantic and surrounding areas) suggests that additional works are necessary. Further investigation about the effects of other internal climate modes (e.g., the Southern Annular Mode [SAM], the Indian Ocean Basin [IOB] mode, the North Tropical Atlantic [NTA]) on evaporation might improve our understanding of the response global hydrological cycle to internal climate variability. Little effort has been made to quantify the influences of external climate factors (i.e., volcanic eruptions, solar variations, and changes in concentration of greenhouse gases) on global evaporation, thus, these analyses might be a subject of forthcoming studies.

**Data availability**

CMIP data can be downloaded from the ESGF website at https://esgf-node. llnl.gov/projects/esgf-llnl/.

**Competing interests**

The authors declare that they have no conflict of interest.

**Author contribution**

TL designed the study. TL performed the data analysis and wrote the manuscript. DHB contributed to the interpretation of results and the writing of manuscript.

**Acknowledgements**

We thank Ryan Teuling, Brecht Martens and Jasper Denissen for valuable comments and suggestions. We acknowledge the World Climate Research Programme's Working Group on Coupled Modeling, which is responsible for CMIP, and we thank the climate modeling groups (listed in Table S1 of this paper) for producing and making available their model output. For CMIP the U.S. Department of Energy's Program for Climate Model Diagnosis and Intercomparison provided coordinating support and led the development of software infrastructure in partnership with the Global Organization for Earth System Science Portals. T. Le is supported by a research grant from Sejong University. This work is supported by the Korea Environmental Industry and Technology Institute (KEITI); the research grant is funded by the Ministry of Environment (grant RE201901084).

[revised manuscript text omitted]

**Figure 5.** Fraction of Earth surface for land and ocean with probability for the absence of Granger causality between climate modes and evaporation less than 0.1 (i.e., *p* value < 0.1). The results are shown for the influence of individual climate mode on annual mean evaporation for periods 1906-2000 (a) and 2006-2100 (b). Fraction areas influenced by ENSO, NAO and IOD are shown in blue, red and yellow bars, respectively. Several fraction areas are close to zero. ENSO = El Niño–Southern Oscillation. NAO = North Atlantic Oscillation. IOD = Indian Ocean Dipole.

MODELS MEAN OF ENSO - EVAPORATION: PERIOD 1906-2000 MINUS PERIOD 2006-2100

[Figure]

MODELS MEAN OF IOD - EVAPORATION: PERIOD 1906-2000 MINUS PERIOD 2006-2100

[Figure]

MODELS MEAN OF NAO - EVAPORATION: PERIOD 1906-2000 MINUS PERIOD 2006-2100

[Figure]

**Figure 6.** Difference of probability for the absence of Granger causality of individual climate mode on annual mean evaporation between periods 1906-2000 and 2006-2100 (i.e., period 1906-2000 minus period 2006-2100). The results are shown for ENSO (a), IOD (b) and NAO (c). Blue shades indicate lower probability of no Granger causality during period 1906-2000 compared to period 2006-2100. Brown shades indicate lower probability of no Granger causality during period 2006-2100 compared to period 1906-2000. ENSO = El Niño–Southern Oscillation. NAO = North Atlantic Oscillation. IOD = Indian Ocean Dipole.

MODELS MEAN OF PREDOMINANCE BETWEEN ENSO, NAO AND IOD - EVAPORATION: PERIOD 1906-2000

[Figure]

MODELS MEAN OF PREDOMINANCE BETWEEN ENSO, NAO AND IOD - EVAPORATION: PERIOD 2006-2100

[Figure]

735  **Figure 7.** The predominance of single climate mode on regional evaporation for periods 1906-2000 (a) and 2006-2100 (b). The predominance of a climate mode at a grid point is defined when the lowest *p* value of all climate modes (see also Figures 1, 2 and 3) at the given grid point is less than 0.25 (i.e., climate modes are unlikely to have no causal effects on evaporation). The predominance of ENSO, NAO and the IOD on evaporation are shown in red, blue and green shades, respectively. ENSO = El Niño–Southern Oscillation. NAO = North Atlantic Oscillation. IOD = Indian Ocean Dipole.